# Zika virus remodelled ER membranes contain proviral factors involved in redox and methylation pathways

Solène Denolly[1], Alexey Stukalov [2], Uladzimir Barayeu [3,4], Alina N. Rosinski [1], Paraskevi Kritsiligkou [3], Sebastian Joecks[1], Tobias P. Dick [3,4], Andreas Pichlmair [2,5] & Ralf Bartenschlager [1,6] ✉

Zika virus (ZIKV) has emerged as a global health issue, yet neither antiviral therapy nor a vaccine are available. ZIKV is an enveloped RNA virus, replicating in the cytoplasm in close association with ER membranes. Here, we isolate ER membranes from ZIKV-infected cells and determine their proteome. Forty-six host cell factors are enriched in ZIKV remodeled membranes, several of these having a role in redox and methylation pathways. Four proteins are characterized in detail: thioredoxin reductase 1 (TXNRD1) contributing to folding of disulfide bond containing proteins and modulating ZIKV secretion; aldo-keto reductase family 1 member C3 (AKR1C3), regulating capsid protein abundance and thus, ZIKV assembly; biliverdin reductase B (BLVRB) involved in ZIKV induced lipid peroxidation and increasing stability of viral transmembrane proteins; adenosylhomocysteinase (AHCY) indirectly promoting $m^6A$ methylation of ZIKV RNA by decreasing the level of S- adenosyl homocysteine and thus, immune evasion. These results highlight the involvement of redox and methylation enzymes in the ZIKV life cycle and their accumulation at virally remodeled ER membranes.

Zika virus (ZIKV), a mosquito-borne flavivirus belonging to the *Flaviviridae* family, was first isolated from an infected captive, sentinel rhesus monkey in the Zika forest in Uganda[1]. A few years later, it was found that ZIKV can also infect humans. We now know that most infections are asymptomatic or cause only mild disease[2], but the last outbreaks in French Polynesia and Brazil revealed that ZIKV infection can cause congenital microcephaly and neurological disorders[3,4]. In 2022, the Pan American Health Organization reported a total of 40,249 confirmed Zika cases in the Americas. Thus far, neither antiviral therapies nor prophylactic vaccines for ZIKV are available[5,6].

After ZIKV entry into the host cell, the positive-strand RNA genome is released into the cytoplasm and translated giving rise to a polyprotein that is cleaved into ten proteins: three structural proteins (capsid, precursor membrane (prM) and envelope (E)) and seven non-structural proteins (NS) required for viral RNA replication (NS1, NS2A, NS2B, NS3, NS4A, NS4B, and NS5)[7]. In the infected cell, ZIKV induces invaginations of the endoplasmic reticulum (ER) membrane, creating clusters of vesicles called vesicle packets, which are presumed sites of viral RNA replication[8,9]. Newly synthesized RNA genomes associate with multiple copies of the capsid protein, thus forming a nucleocapsid that is enveloped by budding into the ER lumen at sites containing multiple copies of prM and E proteins. Resulting immature virions are transported to the Golgi where prM is cleaved to allow conformational changes of

[1]Heidelberg University, Medical Faculty Heidelberg, Department of Infectious Diseases, Molecular Virology, Center for Integrative Infectious Disease Research, 69120 Heidelberg, Germany. [2]Technical University of Munich, School of Medicine, Institute of Virology, 81675 Munich, Germany. [3]Division of Redox Regulation, German Cancer Research Center (DKFZ), DKFZ-ZMBH Alliance, Heidelberg, Germany. [4]Faculty of Biosciences, Heidelberg University, Heidelberg, Germany. [5]German Center for Infection Research (DZIF), Munich Partner Site, Munich, Germany. [6]Division Virus-Associated Carcinogenesis, German Cancer Research Center (DKFZ), Heidelberg, Germany. ✉e-mail: ralf.bartenschlager@med.uni-heidelberg.de

E rendering virions infectious. These are released from the cell by exocytosis.

Besides its key role for ZIKV replication and assembly as a source of membranes, the ER is an essential compartment for controlling cellular lipid homeostasis as well as protein folding and maturation. ZIKV infection was shown to induce ER stress by triggering the unfolded protein response[10,11] and to cause oxidative stress and redox imbalance[12–15]. The induction of this redox alteration appears to be proviral and it is assumed that altered redox state might affect ZIKV protein folding[15] and viral replication[12]. However, the precise mechanisms are poorly defined.

In this work, we purify ER membranes from ZIKV-infected cells in order to identify host cell factors enriched or depleted in this compartment as a result of viral infection. Upon phenotypic validation, we identify 4 enzymes involved in redox and methylation pathways and affecting different steps of the viral life cycle. For each of these factors we characterize the mechanism underlying its proviral function, including protein stability, lipid peroxidation and RNA modification to escape innate immune sensing.

## Results

### Purification of ER membranes from ZIKV-infected cells

Aiming at identifying host cell factors involved in the ZIKV life cycle, we isolated ER membranes from infected A549 cells expressing HA-tagged calnexin (CNX-HA). We chose this approach because the viral replicase is most likely associated with vesicle packets that, in turn, are invaginations of the ER membrane. Thus, viral proteins in these packets might be only poorly accessible to affinity purification as long as the ER membrane is kept intact. To overcome this problem, we chose CNX as a host cell protein being enriched in ZIKV-induced ER membrane clusters (Fig. 1a) and fused the HA-tag to the C-terminus of CNX residing in the cytoplasm. For purification, we used a combination of membrane enrichment via density gradient centrifugation and subsequent HA-affinity capture (Fig. 1b) similar to the protocol we previously used to purify hepatitis C virus induced double membrane vesicles[16]. As controls we employed Mock-infected CNX-HA expressing cells and naïve cells with or without ZIKV infection. Fractions harvested after ultracentrifugation revealed enrichment of NS3, NS4B and viral RNA (total and (-) strand, the latter being only produced during viral replication and therefore serving as the most reliable marker for ZIKV replication sites); in the same fractions the ER marker CNX was enriched whereas cytosolic proteins such as actin were depleted (Fig. 1c–e). For further purification, peak fractions (5–8) were pooled and subjected to HA affinity capture. In this way, we specifically enriched the viral proteins, along with CNX-HA, and (-) strand RNA (Fig. 1f–h).

### Proteins involved in redox reactions are enriched at ZIKV remodeled ER membranes

Eluates of the HA-affinity purification were analyzed by mass spectrometry in order to identify host cell factors enriched or depleted at ER membranes isolated from ZIKV-infected cells, relative to non-infected cells. We detected 46 enriched proteins, including four viral proteins, and eight depleted proteins (Fig. 2a, Supplementary Figure 1 and Supplementary Data 1). Interestingly, we found a strong enrichment of proteins having oxidoreductase activity (Fig. 2b): TXNRD1, PGK1, ALDH3A1, SCD, BLVRB, TXN, SOD1, ALDH1A1, AKR1B1, AKR1B10, AKR1C3 (Supplementary Data 1).

To evaluate the role of identified proteins in the ZIKV life cycle, we conducted a siRNA screen using the experimental layout depicted in Fig. 2c. Values obtained with cells that had been transfected with a non-targeting siRNA pool were used for normalization. After exclusion of those genes where the knock-down (KD) turned out to be cytotoxic, we selected 13 host cell factors having the strongest impact on virus entry and replication (1st round of infection) and virus production (2nd round

of infection) (Fig. 2d). These factors were subjected to a validation screen in the same way by measuring cell viability, intracellular viral RNA as well as extracellular virus titers. In all cases, KD was highly efficient as determined by RT-qPCR (Fig. 3a). For two genes, stearoyl-CoA desaturase (SCD) and triosephosphate isomerase 1 (TIPI1), we observed cytotoxicity and therefore excluded them from further analysis (Fig. 3b). Amongst the other factors, we found that KD of thioredoxin reductase 1 (TXNRD1) and aldo-keto reductase family 1 member C3 (AKR1C3) did not alter the level of intracellular viral RNA, but reduced the production of extracellular infectious virus (Fig. 3c, d, respectively), suggesting that these factors positively regulate a late stage of the ZIKV life cycle. In contrast, KD of biliverdin reductase B (BLVRB) and adenosylhomocysteinase (AHCY) reduced both intracellular viral RNA and extracellular virus titer (Fig. 3c, d), arguing that these factors contribute to an early stage of the viral life cycle. All the other tested factors had no overt effect on the ZIKV life cycle.

A common denominator of the four validated top hits is their involvement in cellular redox homeostasis and/or metabolic interconversion. Therefore, with the aim to decipher the mechanism by which these factors promote the ZIKV life cycle, we analyzed them in further detail. As a first step, we determined changes of the subcellular localization of the four selected host cell factors, relative to CNX-HA, in ZIKV-infected cells (Fig. 3e). While in non-infected cells, these cellular proteins were widely distributed throughout the cytoplasm, their distribution changed upon ZIKV infection to varying extents, in parallel to the altered subcellular distribution of CNX-HA, consistent with these proteins being associated with ZIKV-remodeled ER membranes. Next, we determined the colocalization of the four host cell factors with viral double-stranded (ds)RNA, a ZIKV replication intermediate that can be detected by immunofluorescence in infected cells. Consistent with the CNX-HA based IF data, we observed a strong colocalization of BLVRB, AHCY and TXNRD1 with dsRNA (Fig. 3f, g), but not for AKR1C3. These results argued for different mechanisms by which these proteins promote the ZIKV life cycle and therefore, we analyzed each of these factors individually.

### TXNRD1 regulates secretion of infectious ZIKV particles by modulating the folding of disulfide bond-containing viral proteins

TXNRD1 is the cytosolic thioredoxin reductase 1, a member of the pyridine nucleotide oxidoreductase family. It catalyzes the NADPH-dependent reduction of thioredoxin and other substrates[17]. Consistent with the finding from the validation screen, KD of TXNRD1 neither affected cell viability (Fig. 3b), nor intracellular viral RNA abundance (Fig. 4a), arguing against a contribution of this host cell protein to ZIKV RNA replication. However, we observed an intracellular accumulation of infectious virus particles in TXNRD1 KD cells (Fig. 4b) correlating with a decrease of extracellular virus particles as determined by plaque assay and RT-qPCR (Fig. 4c, d). To determine whether virion secretion inhibition by TXRND1 KD is specific to ZIKV particles or a more general phenomenon, we measured secretion of NS1 from infected cells. We found that TXRND1 KD also impaired NS1 secretion arguing for a more general role of this host cell protein in the secretory pathway (Fig. 4e).

It was previously shown that TXRND1 provides reducing equivalents for transport from the cytosol into the lumen of the ER. Reducing power is needed inside the ER to reduce non-native ("wrong") disulfide bonds in maturing secretory and membrane proteins, thus allowing for the "correction" of wrongly formed disulfide bonds[18]. To confirm the relevance of TXNRD1 in promoting the maturation of disulfide bond containing proteins, we monitored the secretion of two cellular proteins in our system, one with and one without disulfide bonds, namely apolipoprotein E (apoE) and alpha-1-anti-trypsin (A1AT), respectively. As expected, we found that TXNRD1 KD impaired apoE, but not A1AT secretion (Fig. 4f, g). Amongst the ZIKV proteins, NS1 and E contain disulfide bonds[15,19,20], prompting us to speculate that TXRND1 might be

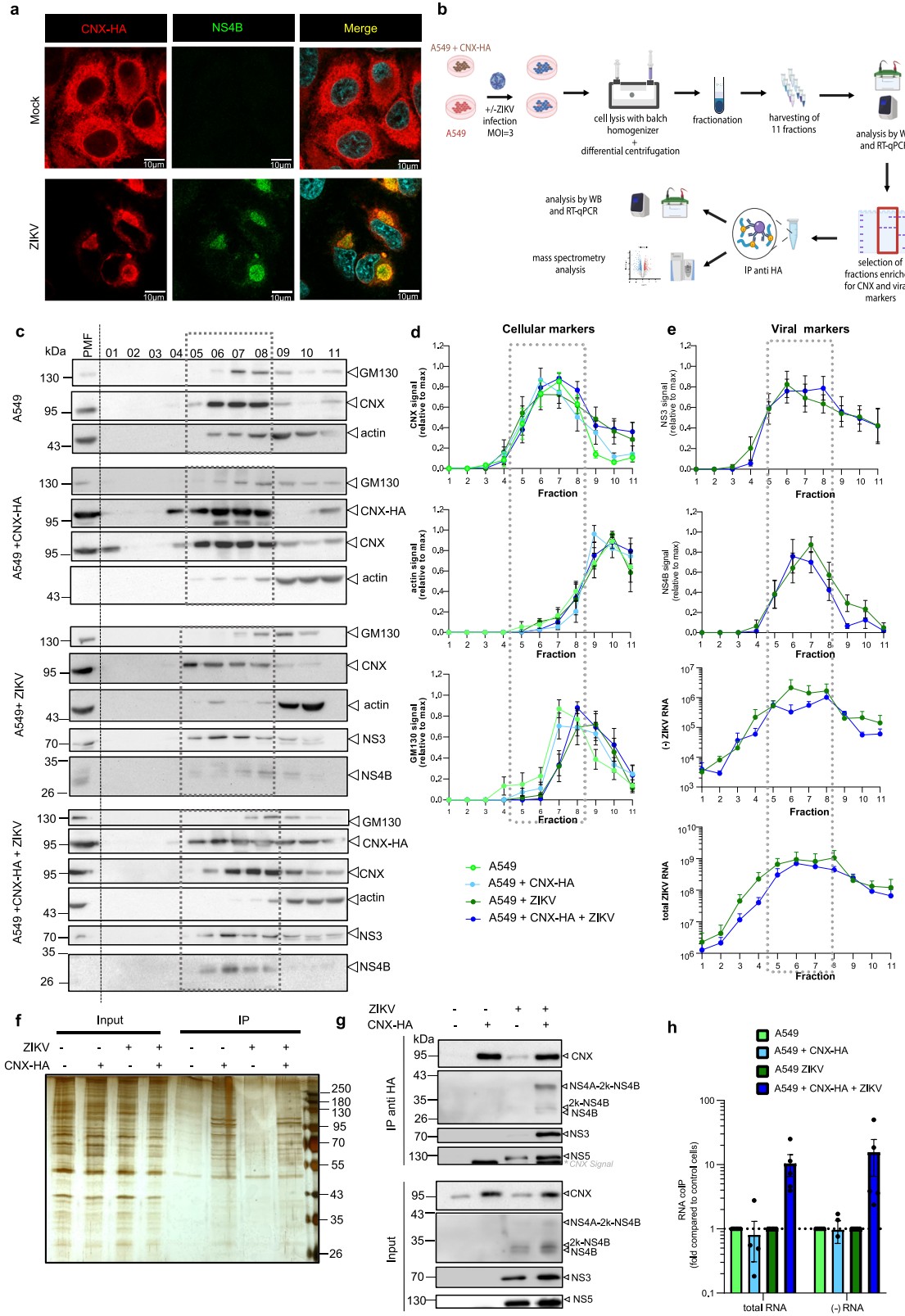

recruited by ZIKV to promote proper folding of these proteins. To test this hypothesis, we analyzed NS1 and E in lysates of TXRND1 KD cells by using SDS-PAGE under both reducing and non-reducing conditions and detection of the proteins by western blot with monoclonal antibodies. For NS1, we used a monoclonal antibody recognizing NS1 under reducing and non-reducing conditions, while for E we used two different antibodies, one recognizing E only under non-reducing

condition[15] (monoclonal antibody 4G2, Supplementary Figure 2a) and one polyclonal antibody allowing detection of E independent of reducing conditions. When E and NS1 were analyzed under reducing conditions, amounts of both proteins were not affected by TXRND1 KD (Fig. 4h, i). However, differences were observed when the proteins were analyzed under non-reducing conditions. In the case of NS1, lower signal was obtained for the sample isolated from TXRND1 KD

**Fig. 1 | Purification of ER membranes from Mock- and ZIKV-infected cells.**
**a** Immunofluorescence analysis of A549 cells stably expressing Calnexin-HA (CNX-HA). Cells were fixed 24 h post infection and stained for CNX-HA and ZIKV NS4B. Scale bars represent 10 μm. Representative image of 3 independent experiments. **b** Purification strategy of ER membranes. A549 WT or CNX-HA cells were mock inoculated or infected with ZIKV (MOI = 3). After 24 h, cells were lysed using detergent-free conditions. After differential centrifugation to remove nuclei and mitochondria, components contained in the post mitochondrial fraction were separated by sucrose gradient centrifugation. Fractions enriched for CNX-HA, NS3, NS4B and viral RNA were pooled and subjected to affinity purification using HA-specific beads. Eluates were analyzed by western blot, RT-qPCR and mass spectrometry. Created with Biorender.com. **c** Representative western blot analysis of fractions collected after sucrose gradient centrifugation. Box indicates those

fractions that were pooled for subsequent analysis. PMF, post-mitochondrial fraction. **d** Quantification of cellular markers in each fraction. CNX, ER membrane marker; actin, cytosolic protein; GM130, Golgi marker. **e** Quantification of ZIKV NS3 and NS4B, total viral RNA and (-) strand ZIKV RNA in gradient fractions. Data in (d) and I are represented as mean ± SEM of 6 independent experiments. **f** Silver stain of proteins contained in pooled fractions (5–8) before (input) and after HA-affinity capture (IP). Representative image of 6 independent experiments. **g** Western blot analysis of pooled fractions before (input) and after anti-HA IP. Representative image of 6 independent experiments. **h** Level of total and (-) strand ZIKV RNA contained in immune-captured samples as determined by RT-qPCR. Values were normalized to those obtained with A549 WT cells. Data are represented as mean ± SEM of 6 independent experiments. Source data are provided as a Source Data file.

cells (Fig. 4h) while for E, the signal was higher (Fig. 4i). Since abundance of both proteins was the same when samples were analyzed under reducing conditions we could exclude differences in protein stability in TXRND1 KD cells. This observation suggested that monoclonal antibody binding is sensitive to and indicative of differences in disulfide bonding for both NS1 and E. While for NS1 the exact epitope is not known, the 4G2 monoclonal antibody used to detect E recognizes a conformational epitope present in the fusion loop of E domain II[21]. Therefore, our results indicated modified accessibility of the epitopes in E, and presumably also in NS1 to the used detection antibodies, with a greater accessibility for E and a decreased one for NS1. No change in protein detection was found for ZIKV NS3 (Supplementary Figure 2b) consistent with the lack of disulfide bonds in this protein. Taken together, these results suggested that TXRND1 KD causes an altered protein conformation, probably due to formation of incorrect disulfide bonds.

To support this assumption, we focused on NS1 and treated ZIKV infected cells with the alkylating agent N-Ethylmaleimide (NEM) prior to cell lysis in order to avoid formation of novel disulfide bonds during cell lysis. Under these conditions, depletion of TXRND1 decreased the amount of mature NS1 (gray triangle in Fig. 4j). Interestingly, depletion of TXRND1 also led to increased accumulation of high molecular weight NS1 aggregates, not entering the gel (white triangle in Fig. 4j), when analyzing the samples under non-reducing conditions. This suggests that NS1 engages in non-native inter-subunit disulfide bonding when TXRND1 is depleted. Taken together, these results suggest that TXRND1 might be involved in the proper folding of NS1 and E by preventing the formation of non-native disulfide bonds.

## BLVRB regulates ZIKV replication by modulating lipid peroxidation
The second redox-related protein promoting the ZIKV replication cycle and enriched in ZIKV remodeled ER membranes was BLVRB, an enzyme converting biliverdin to bilirubin, with the latter being a strong antioxidant involved in counteracting reactive oxygen species (ROS)[22]. In addition, BLVRB has been reported to have a prooxidant function by reducing flavins, which are co-factors for numerous redox enzymes and by reducing complexed ferric iron ($Fe^{3+}$) to ferrous iron ($Fe^{2+}$)[23].

As a first step, we determined the impact of BLVRB KD on the different steps of the ZIKV life cycle. We observed a decrease of intracellular viral RNA and a consistent reduction of intracellular infectivity (Fig. 5a, b), along with reduced amounts of extracellular virus particles as determined by plaque assay and RT-qPCR (Fig. 5c, d). These results argued for a role of BLVRB in ZIKV entry or RNA replication. To determine whether this function requires BLVRB enzymatic activity we tested the effect of two small molecules previously shown to act as BLVRB inhibitors, Phloxine B[24] and Lumichrome[23,24], the latter being an inhibitor of flavin reductases. Both inhibitors lowered intracellular ZIKV RNA levels and reduced the production of infectious virus particles (Supplementary Figure 3a), supporting the notion that the flavin reductase activity of BLVRB is involved in ZIKV replication. The

disproportionally stronger effect of Phloxine B on the production of infectious ZIKV suggests that this compound might have additional effects on assembly, release or infectivity of virus particles.

Recent reports indicate that ZIKV infection induces the intracellular production of ROS[12,15]. Interestingly, we observed a mitigation of ZIKV-induced ROS by BLVRB depletion (Supplementary Figure 3b), suggesting that BLVRB might play a role in this process. Possessing ferric iron reductase activity, BLVRB may drive Fenton chemistry and the initiation of radical chain reactions. We therefore investigated lipid peroxidation (LPO), which can be induced by ROS and which leads to membrane damage and eventually cell death[25]. LPO is mediated by free radicals attacking lipids that contain carbon-carbon double bonds, especially polyunsaturated fatty acids (PUFAs)[25]. To first assess whether ZIKV indeed induced LPO, we measured the product of LPO-induced lipid aldehydes in ZIKV-infected cells by using the fluorescent dye (7-(diethylamino)-coumarin-3-carbohydrazide (Fig. 5e)[26,27]. As controls, we treated the cells with RSL3[28], an inducer of LPO, and Liproxstatin-1[29], an inhibitor of LPO. Interestingly, while the controls increased or reduced LPO as expected, ZIKV infection significantly enhanced LPO 8 h after infection (Fig. 5e, f). Next, we wondered whether BLVRB might play a role in this reaction and repeated the experiment, but using cells transfected with NT- or BLVRB-specific siRNAs. While in mock cells BLVRB KD led to an increase of LPO, ZIKV-induced LPO was mitigated by BLVRB depletion (Fig. 5g, h). To determine if this reduced LPO induction was responsible for the decreased ZIKV replication observed in BLVRB KD cells, we treated the cells with RSL3 in order to chemically induce LPO (Fig. 5i). Interestingly, while the RSL3 treatment caused no significant increase of ZIKV RNA replication in NT-cells, it restored ZIKV intracellular RNA levels and the production of infectious ZIKV particles in BLVRB KD cells. In a complementary approach we treated ZIKV infected cells with the general antioxidant N-acetylcysteine (NAC). Interestingly, NAC treatment decreased ZIKV intracellular RNA and production of infectious particles in NT-siRNA transfected control cells, but had no additional effect in BLVRB KD cells (Fig. 5j). These results suggest that BLVRB acts as a proviral factor by promoting the induction of LPO.

To corroborate this assumption, we treated ZIKV-infected cells with various concentrations of Liproxstatin-1 and observed a dose-dependent inhibition of infectious virus production in two different cell systems, A549 and Huh7-derived cells, treated with non-cytotoxic concentrations of the LPO inhibitor at 1 h post-infection (Fig. 6a). Importantly, by using a ZIKV subgenomic replicon system[30] that bypasses virus entry and only measures viral RNA replication, we observed the same dose-dependent inhibition, demonstrating that Liproxstatin-1 impaired ZIKV replication and not virus entry (Fig. 6b).

LPO inhibition might impair ZIKV replication by affecting viral replicase activity or by altering the abundance of viral replicase proteins e.g. by impairing their stability. To address the latter possibility, we employed an expression system designated pIRO-Z, allowing the synthesis of the viral replicase proteins (NS1 to NS5) and vesicle packets formation independent of viral replication[31]. Cells were

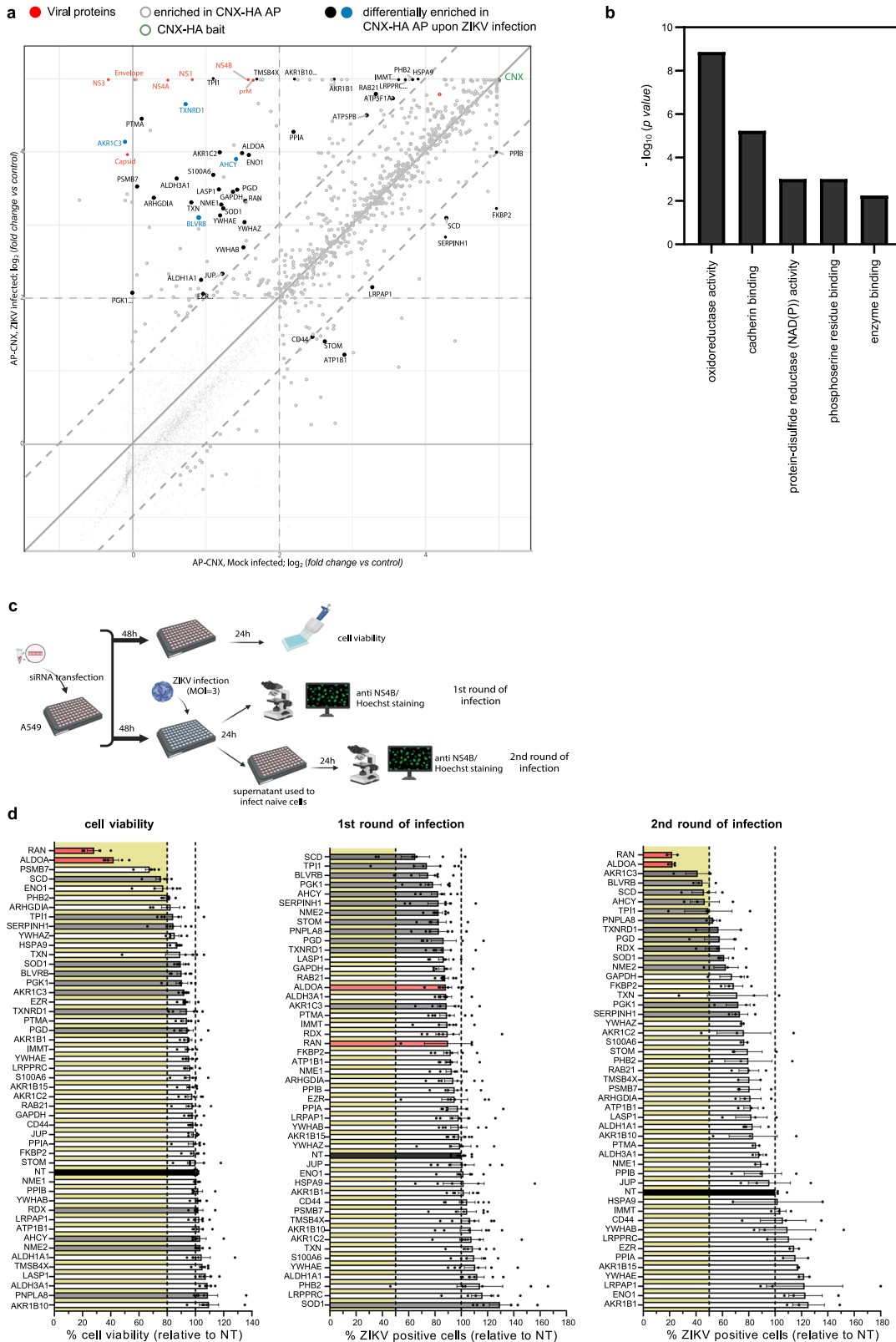

transfected with the pIRO-Z construct, 4 h later treated with Liproxstatin-1 for 15 h and proteins were analyzed by immunoblotting. We found that Liproxstatin-1 treatment of pIRO-Z-transfected cells decreased amounts of NS4B, NS4A and to a lesser extent NS2B, whereas no changes were found with NS1, NS3 and NS5 (Fig. 6c). We noticed that all affected ZIKV proteins contain at least one transmembrane domain whereas the non-affected proteins do not (since

there is no NS2A-specific antibody available to us, this protein could not be analyzed). Since all viral proteins originated from a polyprotein precursor and since the level of NS5, the C-terminal cleavage product, was not affected by Liproxstatin-1 treatment, we could exclude an impairment of RNA translation. Moreover, we did not observe any precursor form of NS4B (Supplementary Figure 3c), arguing against a defect in polyprotein processing.

**Fig. 2 | Identification of host cell factors enriched at ZIKV remodeled ER membranes and their role in the viral life cycle. a** Comparison of protein enrichment upon CNX-HA affinity purification in Mock-infected A549 cells (X axis) and ZIKV-infected A549 cells (Y axis). Proteins significantly enriched upon CNX-HA affinity purification (AP-CNX) in either infection condition ($\log_2$ *fold-change* $\geq 2$, p-value $\leq$ 1E-3 (based on Bayesian linear model; two-sided, unadjusted; $n = 6$ independent biological batches)) are shown as larger dots. Proteins that are also differentially enriched between ZIKV-infected and Mock-infected experiments ($|\log_2$ *fold-change*$| \geq 1$, p-value $\leq$ 1E-3 are (based on Bayesian linear model; two-sided, unadjusted; n = 6)) are depicted as filled circles (viral proteins – red; proteins characterized further – blue). See Methods sections for more detail of the statistical analysis. For a complete list of proteins, see Supplementary Data 1. **b** Top 5 GO enrichment score [-$\log_{10}$ (*P-value*)] analysis of proteins identified in **a**. Fisher's one-tailed test. **c** siRNA screen strategy. A549 cells were reverse transfected with siRNAs

and 48 h later, cells were infected with ZIKV (MOI = 3). One plate was left untreated for cell viability measurement 72 h post transfection while the other plate was harvested 24 h post infection and stained for NS4B to assess infection efficiency. Supernatants were harvested, used to infect naïve A549 cells and 24 h later, cells were stained for NS4B to assess infection efficiency. Created with Biorender.com. **d** Cell viability (left), percentage of ZIKV-positive cells after the first round of infection (middle) and percentage of ZIKV-positive cells after the second round of infection (right). All values were normalized to NT siRNA transfected cells (black bar). Orange bars represent host cell factors where KD caused cytotoxicity while grey bars represent host cell factors selected for further analysis. Data are represented as mean ± SEM of 3 independent experiments conducted in duplicates. Dotted lines indicate 80% and 100% for the cell viability graph (left) and 50% and 100% in the other two graphs. Source data are provided as a Source Data file.

Next, we wondered whether the decreased abundance of the ZIKV transmembrane proteins was due to increased degradation and added the proteasome inhibitor MG132 to pIRO-Z transfected cells. Interestingly, MG132 prevented Liproxstatin-1 induced degradation of NS4A and NS4B (Supplementary Figure 3d). To further corroborate that BLVRB impairs ZIKV replication by modulating stability of viral transmembrane proteins, we expressed the pIRO-Z construct in NT and BLVRB KD cells. Also under these conditions, we observed decreased amounts of NS4A, NS4B and to a lesser extent NS2B, whereas no changes were found for NS1, NS3 and NS5, thus phenocopying the Liproxstatin-1 treatment of the cells (Fig. 6d). Taken together, these results suggest that BLVRB acts as a proviral factor by promoting the induction of LPO, which increases the stability of ZIKV replicase proteins containing transmembrane domains.

### AKR1C3 regulates ZIKV assembly by modulating capsid abundance

The third redox-related protein associated with ZIKV remodeled ER membranes and analyzed here was AKR1C3. This protein belongs to a large superfamily of enzymes that utilize NADH and NADPH as cofactors to catalyze the conversion of aldehydes and ketones to their corresponding alcohols. Elevated AKR1C3 abundance in cancer cells was shown to prevent ferroptosis by reducing the end products of LPO to non-toxic, lipid-derived alcohols[32]. Having found AKR1C3 enriched in ER-membranes of ZIKV-infected cells, we corroborated the KD phenotype (Fig. 3) and found that AKR1C3 depletion did not alter intracellular viral RNA level (Fig. 7a), but reduced the titer of intra- and extracellular virus particles (Fig. 7b–d). This was not due to lower infectivity of released virions because amounts of extracellular viral RNA, a surrogate for virus particles, was comparably reduced (compare Fig. 7c with 7d). Therefore, we hypothesized that AKR1C3 depletion might impair the assembly of infectious virus particles e.g. by reducing the amounts of the viral structural proteins. To address this assumption, we quantified ZIKV structural proteins (capsid, prM and E) in AKR1C3 KD cells. We observed an ~40% decrease of capsid abundance (Fig. 7e), while the amounts of E and prM/M were not affected (Fig. 7f and g, respectively), suggesting that AKR1C3 might regulate ZIKV assembly by regulating capsid amounts.

To determine if capsid stability was regulated by AKR1C3 and more particularly, by which pathway, we treated AKR1C3 KD cells with either MG132, to inhibit the proteasome, or Bafilomycin A1 (Baf A1) to inhibit lysosomal degradation. Cells were infected with ZIKV and 8 h later treated with either drug or DMSO for 16 h. Thereafter, protein amounts were quantified by western blot. While MG132 treatment did not restore capsid levels, Baf A1 treatment did (Fig. 7h), suggesting that AKR1C3 prevented capsid degradation via the lysosomal pathway. To complement this observation, we determined the subcellular localization of capsid, along with the lysosomal marker Lamp1, in NT vs. AKR1C3 KD cells. We observed a higher proportion of Lamp1 signal overlapping with capsid signal in AKR1C3 KD cells, suggesting a higher

proportion of capsid in the lysosomal compartment of cells depleted for AKR1C3 (Fig. 7i). Taken together, these data suggest that AKR1C3 is involved in the ZIKV life cycle by regulating capsid degradation via the lysosomal pathway.

### AHCY regulates ZIKV replication by m⁶A modification of ZIKV RNA

The fourth protein we had identified as enriched at ER membranes of ZIKV-infected cells was the metabolic enzyme AHCY, which is the only enzyme capable to catalyze the conversion of S-adenosylhomocysteine (SAH) to adenosine and L-homocysteine[33]. Because of this activity, AHCY is thought to facilitate methyltransfer reactions by removing excessive amounts of SAH. At first, we determined which step of the ZIKV replication cycle AHCY contributes to. We found that AHCY KD reduced intracellular viral RNA abundance (Fig. 8a), correlating with reduced amounts of intra- and extracellular virus particles (Fig. 8b–d). These results suggested that AHCY might promote ZIKV RNA replication or entry.

It was previously shown that AHCY inhibition reduced the level of methylated RNAs in mouse macrophages and osteosarcoma cells, including RNAs containing N6-methyladenosine (m⁶A)[34]. As ZIKV RNA is also m⁶A methylated[35], we hypothesized that AHCY might be important for this modification of the viral RNA. We subjected the same amount of RNA isolated from ZIKV-infected NT and AHCY KD cells to immunoprecipitation (IP) using an m⁶A-specific antibody and amounts of captured ZIKV RNA were quantified by RT-qPCR. Relative to the level of ZIKV RNA detected in the eluate of NT KD cells, we detected significantly less viral RNA in the eluate from AHCY KD cells (Fig. 8e), arguing that AHCY might be involved in m⁶A methylation of ZIKV RNA. In addition, we used the same immune-capture approach to determine the level of m⁶A modification of two cellular transcripts, which were shown to be m⁶A methylated independent of ZIKV infection, β-actin (ACTB) and Ras GTPase-activating protein-binding protein 2 (G3BP2). While we observed no quantitative change in m⁶A modification of ACTB mRNA, amount of m⁶A modified G3BP2 mRNA was significantly reduced in AHCY KD cells (Fig. 8e), suggesting that AHCY might also regulate m⁶A methylation of some cellular RNAs.

It has been shown that the m⁶A modification reduces RNA recognition by innate immune sensors, thus playing an important role in immune evasion[36,37]. Since the experiments described so far were conducted in A549 cells, which are interferon competent, we hypothesized that the decrease of m⁶A methylation of ZIKV RNA upon AHCY KD might be due to enhanced viral RNA sensing, thus increasing activation of antiviral pathways. We therefore tested the effect of AHCY KD on ZIKV replication in A549 cells having a knock-out of the essential adaptor molecule MAVS (mitochondrial antiviral-signaling protein). These cells, designated A549-MAVS KO[38], do not induce an interferon response upon activation of RIG-I, MDA5 and LGP2 allowing us to evaluate whether m⁶A modification of ZIKV RNA indeed supports

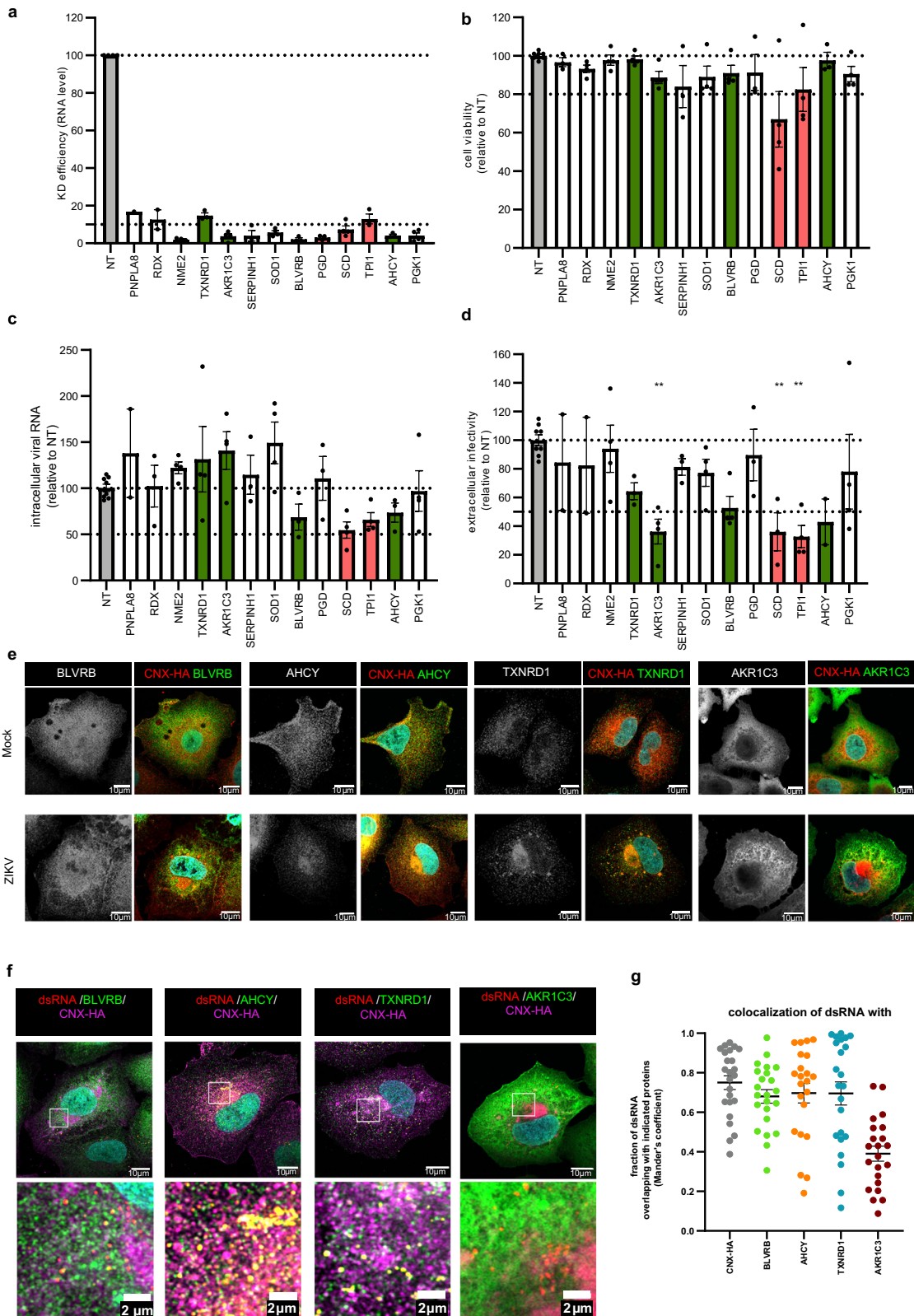

immune evasion. While KD of AHCY was comparable between A549 and A549-MAVS KO cells (Fig. 8f), we could only detect an effect on ZIKV replication by AHCY KD in A549 WT, but not in A549-MAVS KO cells (Fig. 8g and Supplementary Figure 4a). These results indicated that the negative impact of AHCY KD on ZIKV replication was due to elevated induction of an IFN response. Consistently, we observed a higher IFNλ1 mRNA level in ZIKV-infected AHCY KD cells than in NT-KD

control cells (Fig. 8h). As expected, no IFNλ1 activation was found in MAVS KO cells.

To further link the AHCY KD phenotype to a decrease of m6A modification of ZIKV RNA, we targeted the catalytic subunit METTL3 of the N6-adenosine-methyltransferase complex using the inhibitor STM2457. Notably, inhibition of METTL3 impaired ZIKV replication only in A549, but not in A549-MAVS KO cells (Fig. 8i and

**Fig. 3 | TXNRD1, AKR1C3, BLVRB and AHCY are host dependency factors for ZIKV.** A549 cells were reverse transfected with given siRNAs and 48 h later, cells were infected with ZIKV (MOI = 3). One plate was left untreated for cell viability measurement 72 h post transfection while the other plate was harvested 24 h post infection for RNA extraction followed by analysis by RT-qPCR. Supernatants were harvested and used to measure production of infectious virus particles. **a** Knock-down efficiency as assessed by RT-qPCR. Dotted lines serve for orientation. **b** Cell viability as determined by CellTiter Glo assay. Knock-down reducing cell viability to below 80% (lower dotted line) was considered cytotoxic and not considered for further analysis. **c** Level of intracellular ZIKV RNA as determined by RT-qPCR. Values were normalized to siNT-transfected cells. Dotted lines serve for orientation. **d** Extracellular ZIKV titers as determined by plaque assay. Values were normalized to siNT-transfected cells. Dotted lines serve for orientation. $p = 0.0029$ for AKR1C3, $p = 0.0085$ for SCD and $p = 0.0015$ for TPI1. For panels **a**–**d**, data are represented as mean ± SEM, $n = 4$ independent experiments (NT condition was done in duplicate).

Red bars indicate genes where KD caused cytotoxicity; green bars indicate hits analyzed further in this study. For panels **c**, **d**, statistical significance was assessed by one-way ANOVA with Dunnett's multiple comparisons test. **e** Immunofluorescence analysis of A549/CNX-HA cells infected or not (Mock) with ZIKV (MOI = 3) for 16 h and stained for HA and a given host cell factor (red and green in merge panels, respectively). Scale bars represent 10 μm. Representative image of 3 independent experiments. **f** Immunofluorescence of A549/CNX-HA cells infected with ZIKV (MOI = 3) for 16 h and stained for HA (magenta), dsRNA (red) and a given host cell factor (green). Scale bars represent 10 μm and 2 μm for panels and zooms respectively. **g** Degree of colocalization between dsRNA and a given host cell factor or CNX-HA as determined with Manders' correlation coefficient of dsRNA signal overlapping with indicated proteins. Data are represented as mean ± SEM of 3 independent experiments with 8 cells/experiment. Source data are provided as a Source Data file.

Supplementary Figure 4b), consistent with the AHCY KD phenotype. To corroborate this result, we treated naïve A549 cells or A549-MAVS KO cells with SAH that should cause endproduct inhibition of methyltransferases and thereby mimic the absence of AHCY, or with the AHCY inhibitor DZNep[39]. Both treatments decreased ZIKV replication and virus production in A549 control cells whereas this effect was lost in A549-MAVS KO cells (Fig. 8j, k and Supplementary Figure 4c, d).Taken together, our data suggest that AHCY is important for the m[6]A modification of ZIKV RNA, which is required to reduce the activation of innate sensing mounting a MAVS-mediated interferon response.

## TXNRD1, AKR1C3 and BLVRB are also involved in DENV and YFV life cycle

To find out whether the four factors recruited to ZIKV remodeled ER membranes and promoting the viral life cycle also play a role in the life cycle of other flaviviruses, we conducted KD experiments with dengue virus (DENV) that is closely related to ZIKV, and the more distantly related yellow fever virus (YFV). Interestingly, we could show that the KD of the four factors (Supplementary Figure 5a) also impaired the DENV life cycle as determined by quantifying the amount of infectious virus particles (Supplementary Figure 5b). In the case of YFV, KD of TXNRD1, AKR1C3 and BLVRB suppressed YFV production, whereas depletion of AHCY had no effect (Supplementary Figure 5c). This result suggested that TXNRD1, AKR1C3 and BLVRB might play a more general role for the flavivirus life cycle.

In conclusion, our study reveals the recruitment of several host cell enzymes involved in redox and methylation processes, to ZIKV remodeled ER membranes. Thus, ZIKV replication and redox pathways are tightly linked and required to promote different steps of the viral replication cycle including stability and folding of viral proteins, lipid peroxidation and modification of viral RNA, the latter being necessary to allow immune evasion (Fig. 9). The host cell factors reported here might play a more general role in the flavivirus life cycle beyond ZIKV.

## Discussion

In this study, we determined the proteome of ZIKV-remodeled ER membranes in infected cells. We found strong enrichment of metabolic enzymes and more specifically, oxidoreductases, consistent with previous data indicating that ZIKV infection induces oxidative stress[12,14,15,40]. We also show that ZIKV infection induces lipid peroxidation (LPO), which appears to regulate the stability of viral replicase proteins anchored within the ER membrane via transmembrane domains. Lipid peroxides produced upon oxidative stress preferentially form on polyunsaturated fatty acid (PUFA) chains within membranes. Interestingly, the ER is rich PUFAs and was identified as one of the key sites of LPO[41]. LPO alters membrane fluidity and permeability, which could explain why LPO affects the abundance of transmembrane domain-anchored proteins. Interestingly, in the case

of the hepatitis C virus (HCV), LPO was proposed to negatively regulate viral replication by altering the conformation of the NS3-4A protease/helicase and the NS5B RNA-dependent RNA polymerase, thus facilitating HCV persistence[42]. In contrast to HCV, we observed the opposite effect for ZIKV, i.e. enhancement of viral replication by LPO. Although we do not know the exact mechanism by which impaired LPO promotes proteasomal degradation of transmembrane-containing ZIKV replicase proteins, we note that replication enhancement by LPO might not be unique to ZIKV as several recent reports indicate a proviral role of LPO also for other virus families[43,44]. As the exact lipid composition of vesicle packets is unknown, we can only speculate on how LPO may support ZIKV replication. For instance, LPO might create a membrane environment favorable for proper insertion of viral transmembrane proteins. It is also possible that viral proteins are post-translationally modified by reactive lipid species which could regulate protein stability and/or function, as reported for other proteins[45]. A key factor in this pathway appears to be BLVRB that unlike biliverdin reductase A (BLVRA) can act as a flavin reductase[46], and also exhibits a ferric reductase activity[23]. Generated ferrous iron ($Fe^{2+}$) could then react with $H_2O_2$ via the Fenton reaction resulting in formation of hydroxyl radical (•OH), which could then react with unsaturated lipids, resulting in LPO[47]. We therefore propose that BLVRB promotes LPO by promoting Fenton chemistry or by reducing flavins regulating other cellular redox functions[48]. The induction of this oxidative micro-environment and remodeling of ER membranes by LPO might contribute to the stability of transmembrane domain-containing proteins and thus, enhance ZIKV replication.

Apart from BLVRB, we also found TXRND1 enriched at ZIKV remodeled ER membranes. Our data suggest that TXRND1 modulates the secretion of infectious ZIKV particles and, as deduced from impaired NS1 secretion, possibly the secretion of disulfide bonds-containing proteins in general (Fig. 4). It was previously shown that TXRND1, via its reductase activity, can contribute to folding and trafficking of low-density lipoprotein receptor (LDL-R)[18] by delivering reducing equivalents to the ER. The availability of reducing equivalents in the ER is needed for the reduction of non-native disulfide bonds that prevent the proper folding of secretory proteins. Interestingly, the ER luminal ZIKV proteins E and NS1 contain multiple disulfide bonds[19,20]. The observation that TXRND1 KD alters the recognition of these proteins by monoclonal antibodies under non-reducing conditions argues for a conformational change when TXRND1 is depleted. Based on our proteome and colocalization data (Figs. 2a, 3f), we speculate that TXRND1 might be recruited in proximity to ZIKV assembly sites to facilitate proper folding of viral proteins like E and NS1 and in this way, their secretion. Since such misfolding of E would have little impact on the amount of infectious intracellular virus particles (Fig. 4b), the primary defect might be virion secretion rather than virion infectivity.

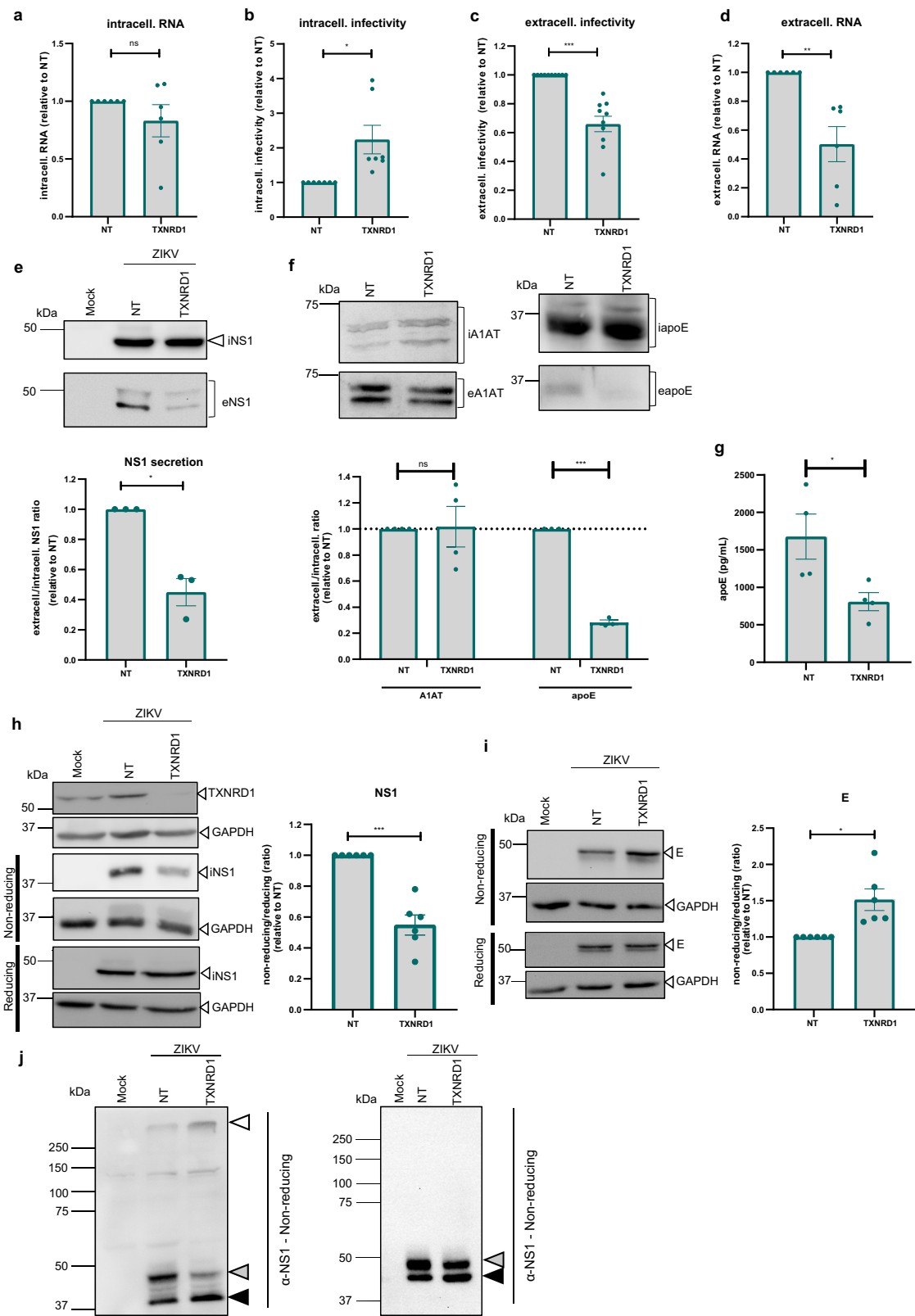

The third host cell protein involved in redox pathways and enriched at ZIKV remodeled ER membranes was AKR1C3. It belongs to the aldo-keto reductase (AKR) superfamily that is characterized by similar structure and the ability to catalyze aldehyde or ketone reduction[49]. Yet, only AKR1C3 was found to play a role in the ZIKV life cycle by modulating the abundance of the viral capsid protein that is degraded by the lysosomal pathway. While the underlying mechanism remains to be determined, it was recently shown that AKR1C3 plays a role in lipophagy, i.e. the degradation of lipid droplets (LDs) via autophagy[50]. Interestingly, LDs have been reported to play a role in the ZIKV life cycle in at least two different ways: First, ZIKV capsid could be observed around LDs[51,52]; second, ZIKV infection induces lipophagy[53]. Therefore, we speculate that AKR1C3 might regulate lipophagy to maintain the pool of LDs containing ZIKV

**Fig. 4 | Role of TXRND1 in ZIKV particle secretion.** A549 cells transfected with TXRND1-targeting or NT siRNAs for 48 h were infected with ZIKV (MOI = 3). Samples were harvested 24 h post infection. **a** Intracellular ZIKV RNA (mean ± SEM; $n = 6$, $p = 0.2809$). **b** ZIKV intracellular infectivity (mean ± SEM; n = 7, $p = 0.0241$). **c** ZIKV extracellular infectivity (mean ± SEM, n = 10, $p = 0.0001$). **d** Amounts of extracellular viral RNA (mean ± SEM, $n = 6$, $p = 0.0095$). **e** Lysates and supernatants of siRNA transfected and ZIKV infected A549 cells analyzed by western blot to detect intra- (i) and extracellular-(e) NS1 (top panel). Signal intensities from 3 independent experiments were normalized to those obtained with NT siRNA transfected cells (bottom panel). mean ± SEM, $p = 0.0258$. **f** Same as **e** for intra- (i) and extracellular (e) A1AT or apoE, respectively (top panel). Signal intensities from 3 (apoE, $p = 0.0009$) or 4 (A1AT, $p = 0.99$) independent experiments were normalized to those obtained with NT siRNA transfected cells (bottom panel). mean ± SEM. For panel **a**–**e** and, statistical significance was assessed by a one-sample t-test (two-tailed). **g** Supernatant of siRNA transfected A549 cells analyzed by ELISA to quantify apoE level. Data represent the mean ± SEM of 4 experiments. $p = 0.0286$ obtained with Mann Whitney test (two-tailed). **h** Lysates of siRNA transfected A549 cells were analyzed by western blot using reducing and non-reducing conditions (left panel). NS1 signals were quantified and normalized to the GAPDH loading control. Shown in the right panel are the ratios of NS1 amounts detected under non-reducing vs. reducing conditions (mean ± SEM, $n = 6$, $p = 0.0009$ obtained with one-sample test (two-tailed). **i** Lysates of A549 cells in **h** were analyzed the same way using two different E-specific antibodies: 4G2 for samples under non-reducing conditions and a polyclonal antiserum for E detection under reducing condition (mean ± SEM, $n = 6$, $p = 0.0183$ obtained with one-sample test (two-tailed). **j** Lysates of siRNA transfected and NEM treated A549 cells were analyzed by NS1-specific western blot using reducing and non-reducing conditions. White triangle, NS1 aggregates gray triangle, mature NS1; black triangle, immature NS1. n = 3 independent experiments. Source data are provided as a Source Data file.

capsid. Alternatively, AKR1C3 might reduce 4-hydroxy-2-nonenal (4-HNE), a cytotoxic aldehyde generated by LPO, into a less toxic metabolite[54]. As 4-HNE can react with proteins to form HNE-protein adducts, it is possible that capsid might react with this aldehyde, thus altering capsid properties, including its stability. In this scenario, AKR1C3 might prevent this reaction by reducing 4-HNE amounts[55].

Finally, we found that AHCY is recruited to ZIKV remodeled ER membranes where it contributes to the m6A modification of ZIKV RNA. M6A nucleosides have been reported to be enriched in ZIKV RNA, but the role of this modification is not fully understood[35], because it is difficult to selectively block methylation of viral RNA without affecting cellular mRNAs. Instead of targeting a direct "writer" or "eraser" of methylation as it was previously done[35], in our study we primarily targeted a protein that promoted methylation by degrading S-adenosylhomocysteine (SAH) that inhibits methyl transferases via a negative feedback-loop because of its structural similarity to S-adenosyl-L-methionine (SAM)[33]. We show that AHCY KD or pharmacological inhibition impaired ZIKV replication only in those cells able to induce a MAVS-dependent interferon response, suggesting that AHCY promotes ZIKV replication by facilitating escape from innate sensors, notably RIG-I reported to be the main sensor of ZIKV infection[56–58]. Thus, by promoting m6A modification of ZIKV RNA, AHCY appears to contribute to the protection of viral RNA against immune receptor recognition as it was previously shown for human metapneumovirus[59].

Interestingly, the four host cell factors characterized here were enriched in purified ER membrane preparations isolated from ZIKV infected cells, arguing for their selective recruitment upon infection. For three of them we confirmed colocalization with dsRNA, indicating close proximity to ZIKV replication/assembly sites. For BLVRB, we suggest that its recruitment to the ER membrane creates a local redox microenvironment conducive to ZIKV replication. Recruitment of AHCY might promote viral RNA methylation by hydrolyzing SAH, potentially accumulating close to replication sites. The recruitment of TXRND1 to sites of replication/assembly appears to be important for promoting correct disulfide bond formation in accumulating ER luminal viral proteins such as E and NS1. The enrichment of AKR1C3 in our preparation is less clear. It might be recruited to ER membranes as a result of infection, but not necessarily directly to ZIKV replication or assembly sites. Instead, it might only indirectly prevent capsid degradation. Further studies will be required to clarify the underlying mechanism.

In conclusion, our data identify several cellular proteins involved in red-ox and methyltransfer reactions contributing to different steps of the ZIKV life cycle. These proteins become enriched at ZIKV remodeled ER membranes, the sites of viral RNA translation, replication and assembly, suggesting that ZIKV hijacks these pathways for its own benefit.

# Methods

## Cells
A549, Huh7-Lunet-T7 and HEK 293 T cells were cultured in Dulbecco's modified Eagle medium (DMEM) complete containing 10% fetal bovine serum (FBS; Sigma-Aldrich), 100 U/mL penicillin, 100 μg/mL streptomycin, and 1% MEM non-essential amino acids (all from Gibco, Life Technologies) and kept at 37 °C with 5% $CO_2$. Stable expression of the T7 RNA polymerase was maintained by culturing the cells in medium containing Zeocin (5 μg/ml; Invitrogen). The C6/36 mosquito cell line was grown in Leibovitz L-15 medium supplemented with 10% FBS, 100 U/mL penicillin and 100 μg/mL streptomycin and kept at 28 °C. A549 cells stably expressing calnexin with a C-terminal HA-tag were generated by lentiviral transduction as described previously[60] and cultured in medium containing 10 μg/ml blasticidin (Invitrogen). A549-MAVS KO cells have been reported recently[56].

## Virus
Stocks of ZIKV (strain H/PF/2013), DENV (New Guinea C isolate) and YFV (17D vaccine strain) were produced by virus amplification in the mosquito cell line C6/36. Virus titers were determined by plaque forming units (PFU) assay in VeroE6 cells as reported earlier[9].

## Reagents
All antibodies and primers used in this study are available in the Supplementary information.

## Purification of ER membranes
A549 or A549-CNX-HA cells were infected with ZIKV (MOI = 3) or Mock-infected for 24 h. Thereafter, cells were washed 3 times, scraped into PBS, pelleted by centrifugation at 700 × g for 5 min at 4 °C and resuspended in 1.5 ml isotonic lysis buffer (ILB; 250 mM sucrose, 10 mM HEPES [pH7.4], 2 mM EDTA, 2 mM Mg-acetate). Cells were lysed by 50 strokes with a balch homogenizer and nuclei and unbroken cells were removed by centrifugation at 700 × g for 5 min at 4 °C. Post-nuclear supernatants were centrifuged at 3000 × g for 10 min at 4 °C to remove mitochondria. Post-mitochondrial fraction (PMF) was layered on top of a discontinuous (30% to 10%) sucrose gradient and centrifuged for 3 h at 100,000 × g using a SW40 rotor (Beckman Coulter, Fullerton, CA). Eleven fractions (32 droplets each) were collected from the bottom of the tube and analyzed for density, protein and ZIKV RNA content. Prior to subsequent HA-specific affinity capture, equal volumes of pooled fractions 5 to 8 were equilibrated to 150 mM NaCl. Incubation with HA-magnetic beads (ThermoFisher Scientific) was carried out for 3 h by continuously inverting the tubes at 4 °C. Beads were washed five times with ILB supplemented with 150 mM NaCl. Bound material was eluted by incubation with 75 μl elution buffer (3% SDS, 50 mM Tris-HCl [pH7.5]), two times, each for 15 min at room temperature (RT). Eluates were analyzed as described in the Results section.

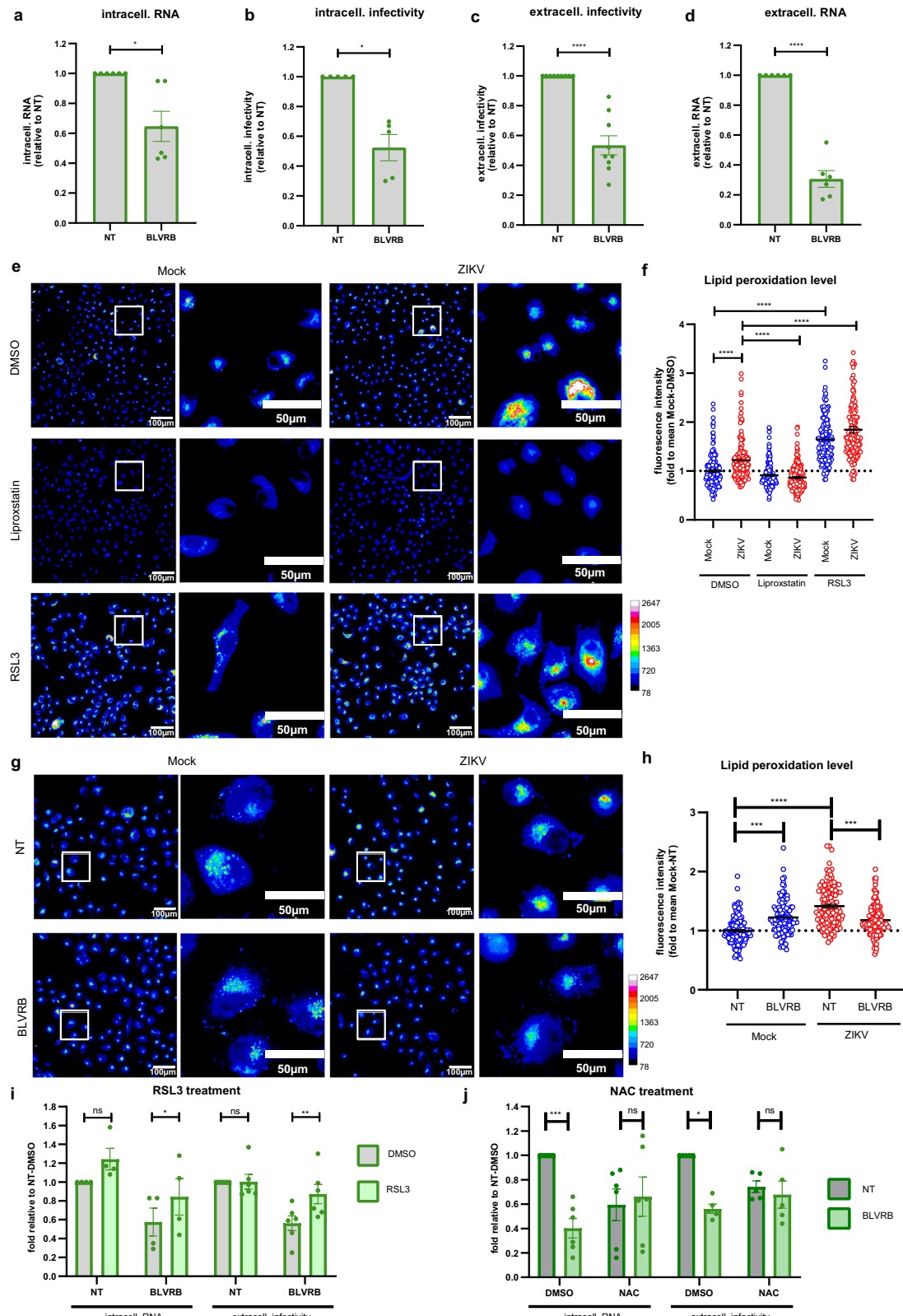

## Mass spectrometry measurements and analysis

For mass spectrometry analysis eluted fractions were pooled, proteins were enriched using SP3 beads and proteins were digested with Trypsin/LysC following a published method[61]. Peptide mixtures were loaded on a 50 cm reverse-phase analytical column (75 μm diameter, 60 °C; ReproSil-Pur C18-AQ 1.9 μm resin; Dr. Maisch), separated using an EASY-nLC 1200 system and directly analyzed on a Q-Exactive HF

mass spectrometer equipped with a nano-electrospray source (all from Thermo Fisher Scientific). The mass spectrometer was operated in positive ionization mode, the spray voltage was set to 2.4 kV, funnel RF level at 60, and heated capillary at 250 °C. Peptides were separated using a 120 min gradient at a flow rate of 300 nl/min, and a binary buffer system consisting of buffer A 0.1% (v/v) formic acid in water, and buffer B 80% (v/v) acetonitrile, 0.1% (v/v) formic acid in water as

**Fig. 5 | BLVRB regulates ZIKV replication by promoting lipid peroxidation.**
A549 cells were transfected with BLVRB-targeting or NT siRNAs for 48 h, followed by ZIKV infection (MOI = 3). Samples were harvested 24 h post infection.
**a** Intracellular ZIKV RNA (mean ± SEM, $n = 6$, $p = 0.0172$). **b** Infectious intracellular infectivity (mean ± SEM, $n = 5$, $p = 0.0057$). **c** Infectious extracellular infectivity (mean ± SEM, $n = 9$, $p < 0.0001$). **d** Amount of extracellular ZIKV RNA (mean ± SEM, $n = 6$, $p < 0.0001$). For panels **a**–**d** one-sample t-test (two-tailed). **e** Lipid peroxidation was determined by staining of Mock and ZIKV infected cells 8 h post infection and treated with Liproxstatin-1 (10 μM) or RSL3 (10 μM) 1 h post infection with 7-(diethylamino)-coumarin-3-carbohydrazide. Scale bars represent 100 μm and 50 μm for panels and zooms respectively. Colors scale represents gray value level. **f** Quantification of fluorescence intensity in **e** (40 cells/experiment; $n = 4$). Values were normalized to those of Mock-DMSO cells (mean ± SEM; one-way ANOVA with Sidak's multiple comparisons test, $p < 0.0001$). **g** Lipid peroxidation in A549 cells transfected with BLVRB-targeting or NT siRNAs for 48 h and infected for 8 h was visualized by using coumarin hydrazide. **h** Quantification of fluorescence intensity in **g** (40 cells/experiment; $n = 3$) Values were normalized to those obtained with Mock-NT cells (mean ± SEM; one-way ANOVA with Sidak's multiple comparisons test, $p = 0.0005$ (Mock NT vs. BLVRB), $p = 0.0002$ (ZIKV NT vs. BLVRB), $p < 0.0001$ (ZIKV NT vs Mock NT)). **i** A549 cells were transfected with given siRNAs for 48 h, followed by ZIKV infection (MOI = 3). Cells were treated with DMSO or RSL3 (1 μM) 1 h post infection and samples were harvested 24 h post-infection. Data were normalized to NT-DMSO controls (mean ± SEM, $n = 4$(RNA), $n = 6$(infectivity); two-way ANOVA with a Sidak's multiple comparisons test (DMSO vs RSL3), $p = 0.0766$ (RNA-NT), $p = 0.0449$ (RNA-BLVRB), $p > 0.9999$ (infectivity NT), $p = 0.0038$ (infectivity BLVRB)). **j** same as **i** with DMSO or NAC (1 mM). Data were normalized to NT-DMSO controls (mean ± SEM, $n = 5$ (infectivity), $n = 6$(RNA); two-way ANOVA with a Sidak's multiple comparisons test (NT vs BLVRB), $p = 0.0002$ (RNA DMSO), $p = 0.9765$ (RNA NAC), $p = 0.015$ (infectivity DMSO), $p = 0.9856$ (infectivity NAC)). Source data are provided as a Source Data file.

follows: 5–30% (95 min), 30–95% (10 min), wash out at 95% for 5 min, readjustment to 5% in 5 min, and keeping at 5% for 5 min. Data-dependent acquisition included repeating cycles of one MS1 full scan (300–1650 m/z, R = 60,000 at 200 m/z) at an ion target of $3 \times 10^6$ with injection time of 20 ms. For MS2 scans the top 15 intense isolated and fragmented peptide precursors (R = 15 000 at 200 m/z, ion target value of $1 \times 10^5$, and maximum injection time of 25 ms) were recorded. Dynamic exclusion, isolation window of the quadrupole, and HCD normalized collision energy were set to 20 s, 1.4 m/z, and 27%, respectively.

Raw MS data files were processed with MaxQuant (version 2.0.3.1) using the default settings for label-free quantification (LFQ). MS Spectra were searched against UniProt human proteome (UP000005640, reference and additional sequences, version Nov-2021) and ZIKV protein sequences defined in A0A024B7W1 UniProt entry.

Data analysis was done in R (version 4.0) and Julia (version 1.6) using in-house scripts[62]. The MaxQuant output files were imported into R with the in-house msimportr R package[63]. Protein groups were redefined using alternative criteria: a group was required to have either 2 unique peptides or at least 25% of its peptides should be unique; similar protein groups with fewer specific peptides were merged together.

To estimate how the intensity of each protein group changes across the samples, we applied Bayesian generalized linear model to the precursor-level data using the msglm package[64] following the methodology described earlier[65]. The linear model (in R GLM package notation) was

$$\log_2 Intensity \sim 1 + treatment*bait + batch,$$

where "treatment" term can take either "Mock" or "ZIKV" value, "bait" term takes either "empty" or "CNX" value, and "batch" specifies the replicate of the experiment. The effects corresponding to the experimental conditions ("treatment" and "bait") were assigned regularized horseshoe+ priors[66], the "batch" effect was assigned normally distributed prior. MS intensity measurement error was considered Laplacian-distributed with parameters Laplace($I$, $\sigma(I)$), where $I$ is the expected signal intensity. The function $\sigma(I)$ that specifies the signal-dependent scale of the measurement error was calibrated using the technical replicate MS data of the instrument. The same technical replicates were used to calibrate the logit-based model of missing MS data (the probability that the MS instrument will fail to identify the protein given its expected abundance in the sample). The model was fit for each protein group individually using the normalized precursor intensities of its specific peptides. The inference was done using rstan[67] (version 2.19): 4000 iterations (2000 warmup + 2000 sampling) of the no-U-turn Markov Chain Monte Carlo were performed in 7 independent chains, every 4th sample was collected into posterior distribution.

To properly assess both the specific enrichment of CNX interactors over the background and their differential binding upon ZIKV infection, 2 normalizations and corresponding model fits were calculated:

- First, the normalization using the proteins found both in empty and CNX-HA APs was performed to equalize the background proteome levels in these APs. This normalization was used to calculate the enrichments of protein groups in CNX-HA AP over empty AP and identify the specific CNX-HA interactors.
- Second, the normalization using the specific CNX-HA interactors was performed to equalize the expression of the bait and ZIKV infection across the batches. This normalization was used to calculate the CNX:ZIKV interaction effect of the model.

To estimate the statistical significance of model effects, the P-values were calculated as the probability that a random sample from the posterior distribution of the effect would be smaller (or larger) than zero. Given the sparsity-inducing property of regularized horseshoe+ priors, no multiple hypothesis testing corrections were applied.

The mass spectrometry proteomics data have been deposited to the ProteomeXchange Consortium via the PRIDE[68] partner repository with the dataset identifier PXD043372.

Gene Ontology (GO) analysis was made with https://biit.cs.ut.ee/gprofiler/gost[69].

### siRNA-based knockdown experiments

For the primary siRNA screen, A549 cells seeded into 96-well plates were reverse transfected with 50 nM of siRNAs targeting specified genes (Dharmacon, complete list and sequence in Supplementary Table 1) by using RNAiMAX Reagent (Thermo Fisher Scientific). For each gene, siRNAs were transfected in duplicate to monitor virus infection and cell viability. After 48 h, cells were infected with ZIKV (MOI = 3) for 1 h. Medium was replaced and after 24 h, cells were washed with PBS and fixed with 4% PFA while supernatants were collected and used to infect naïve A549 cells. Non-infected cells were used to measure cell viability. Twenty four hours post infection of naïve cells, cells were washed with PBS and fixed with 4% PFA. Fixed cells were stained for NS4B and nuclear DNA with Hoechst prior to imaging with a Zeiss Cell Discoverer 7 microscope equipped with a 10x objective. The number of NS4B-positive cells and the total number of cells, assessed with Hoechst staining, were determined by using the ImageJ software package and used to calculate the percentage of NS4B-positive cells.

For validation of primary hits, A549 cells were reverse transfected with 50 nM of siRNA targeting selected genes (Dharmacon) using RNAiMAX Reagent (Thermo Fisher Scientific). After 48 h, cells were infected with ZIKV (MOI = 3), 1 h later medium was exchanged and 24 h post infection, supernatants were collected. After clarification of supernatants by centrifugation at 700 × g for 5 min, virus contained

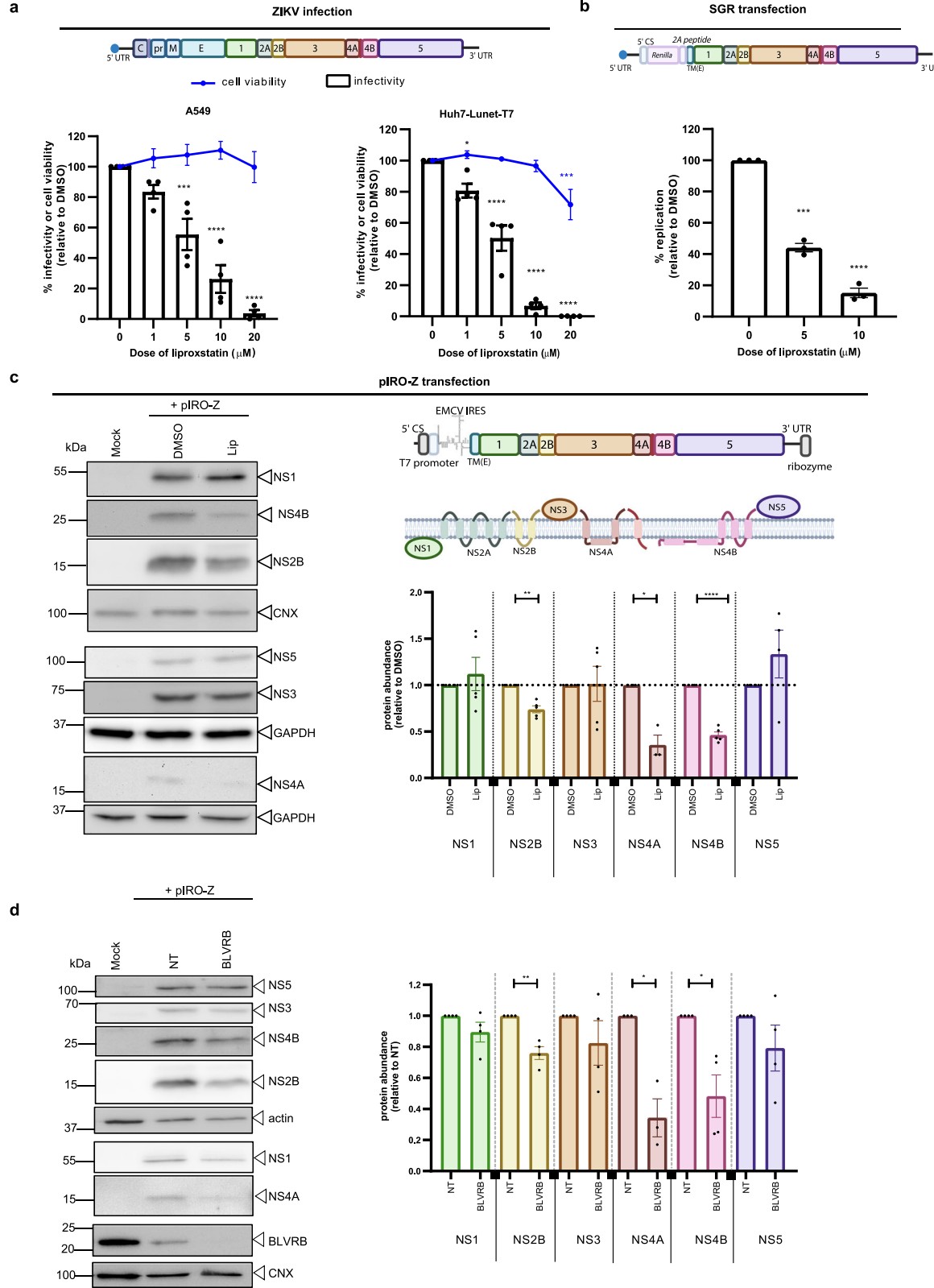

therein was quantified by plaque assay and detection of viral RNA by RT-qPCR. Cells were washed with PBS prior to lysis and detection of viral RNA or cellular mRNA.

### Determination of virus titer by plaque assay

To quantify amounts of infectious extracellular virus, culture supernatants were clarified by centrifugation at $700 \times g$ for 5 min. To determine the amounts of infectious intracellular virus particles, cells were washed with PBS and detached using Versene™ solution (Thermo Fisher Scientific). After centrifugation at $700 \times g$ for 5 min, cells were resuspended in DMEM containing 10% FCS and 15 mM HEPES [pH7.4] and lysed with five freeze-thaw cycles. Cell lysates were centrifuged at $16,000 \times g$ for 10 min and virus contained in supernatants was quantified.

**Fig. 6 | Lipid peroxidation promotes ZIKV RNA replication by increasing the abundance of transmembrane-containing replicase proteins. a** A549 or Huh7-Lunet-T7 cells were infected with ZIKV (top panel) and treated with given doses of Liproxstatin-1 for 24 h. Cell viability (blue) and ZIKV titers (bars) (mean ± SEM, $n = 4$, two-way ANOVA with a Dunnett's multiple comparisons test (comparison to 0); for A549: $p = 0.2503$ (1 µM), $p = 0.0002$ (5 µM), $p < 0.0001$ (10–20 µM); for Huh7-Lunet-T7: $p = 0.0186$ (1 µM), $p < 0.0001$ (5-10-20 µM). **b** Huh7-Lunet-T7 cells stably expressing firefly luciferase were electroporated with RNA of a subgenomic ZIKV replicon encoding renilla luciferase (top panel) and 2 h later treated with Liproxstatin-1 (5 and 10 µM). Cells were harvested 4 h and 24 h post electroporation and replication were determined by dual luciferase assay. Values were normalized to the 4 h-value and to DMSO treated cells (mean ± SEM, $n = 3$, RM one-way ANOVA with Holm-Sidak's multiple comparisons test, $p = 0.0002$ (5 µM), $p < 0.0001$ (10 µM). **c** Huh7-Lunet-T7 cells were transfected with a plasmid encoding ZIKV NS1-NS5 (pIRO-Z; top panel) and treated with DMSO or Liproxstatin-1 (10 µM) 4 h post

transfection. Cells were lysed 19 h post transfection and lysates were analyzed by western blot (left panel). Signal intensities were quantified and normalized to CNX or GAPDH loading control (right panel). Values were normalized to DMSO control (mean ± SEM; $n = 3$ for NS4A, $n = 4$ for NS5 and $n = 5$ for NS1, NS2B, NS3, NS4B; one-sample t-test (two-tailed) for each individual protein, $p = 0.5385$ (NS1), $p = 0.0024$ (NS2B), $p = 0.9444$ (NS3), $p = 0.0264$ (NS4A), $p < 0.0001$ (NS4B), $p = 0.2844$ (NS5)). **d** Huh7-Lunet-T7 cells were transfected with BLVRB-targeting or NT siRNA for 48 h and then transfected with the pIRO-Z plasmid. Cells were lysed 19 h post transfection and lysates were analyzed by western blot (left panel). Signal intensities were quantified and normalized to CNX or actin loading control (right panel). Values were normalized to NT (mean ± SEM; $n = 3$ for NS4A and $n = 4$ for all proteins; one-sample t-test (two-tailed) for each individual protein, $p = 0.1965$ (NS1), $p = 0.001$ (NS2B), $p = 0.3093$ (NS3), $p = 0.0331$ (NS4A), $p = 0.0327$ (NS4B), $p = 0.2552$ (NS5)). Source data are provided as a Source Data file. Images of constructs and proteins were created with Biorender.com.

To measure virus titers, confluent monolayers of VeroE6 cells were inoculated with serial 10-fold dilutions of virus containing supernatants or cell lysates for 1 h at 37 °C. The inoculum was replaced by serum-free MEM (Gibco, Life Technologies) containing 1.5% carboxymethylcellulose (Sigma-Aldrich) and after 4 days for ZIKV, 7 days for DENV and 6 days for YFV, cells were fixed for 1 h at room temperature with formaldehyde directly added to the medium to a final concentration of 5%. Fixed cells were washed extensively with water before 20-min staining with 1% crystal violet dissolved in 10% ethanol. After rinsing with water, the number of plaques was counted and virus titers were calculated[9].

### RNA quantification by RT-qPCR
Cells or culture supernatants were lysed using LBP buffer contained in the NucleoSpin RNA Plus extraction kit (Macherey-Nagel). For samples from cell culture supernatants or obtained after fractionation, exogenous in vitro transcribed HCV RNA was added to the lysate and used as a reference for normalization. Total RNA was isolated using the NucleoSpin RNA Plus extraction kit (Macherey-Nagel) as recommended by the manufacturer. cDNA was generated using the High Capacity cDNA RT kit (ThermoFisher scientific). For negative strand ZIKV RNA, cDNA was generated using SuperScript III First-Strand Synthesis (ThermoFisher scientific) and primer 5′-ggccgtcatggtggcgaataaAGGATCATAGGTGATGAAGAAAAGT-3′, according to[70] and the instruction of the manufacturer. qPCR was performed using the iTaq Universal SYBR green mastermix (Bio-Rad) and primers specified in Supplementary Table 2. Abundance of intracellular ZIKV RNA or cellular mRNA was normalized to HPRT transcript level while abundance of extracellular ZIKV RNA or viral RNA contained in gradient fractions was normalized to the level of exogenously added calibrator (HCV) RNA. All the primers used for qPCR are available in Supplementary Table 2.

### SDS-PAGE and western blot
Cell were washed with PBS and lysed in lysis buffer (20 mM Tris [pH 7.5], 1% Triton X-100, 0.05% sodium dodecyl sulfate, 150 mM NaCl, 0.5% Na-deoxycholate), supplemented with protease/phosphatase inhibitor cocktail (Roche) for 30 min on ice. Nuclei were removed from lysates by centrifugation at $16,000 × g$ for 10 min at 4 °C. For the experiment with NEM, cells were washed with PBS, incubated on ice for 10 min with 100 mM NEM and lysed with PBS containing 1% Triton X-100 and protease/phosphatase inhibitor cocktail (Roche) for 30 min on ice. Samples were then processed as described for non-NEM treated ones. Clarified lysates were mixed with Laemmli buffer (3% SDS, 30% glycerol, 16.3 mM Tris-HCl [pH6.8], bromophenol blue 0.01 g/mL), containing or not 1.3% β-mercaptoethanol to dissolve disulfide bonds or to retain them (reducing or non-reducing conditions, respectively) and incubated for 5 min at 95 °C. Proteins were separated by electrophoresis into SDS - polyacrylamide gels and transferred to a PVDF membrane. Membranes were blocked with PBS-Tween 0.1%,

supplemented with 5% milk. Primary antibodies (see Supplementary Table 3 for detail) were diluted in PBS-Tween 0.1% containing 5% milk and membranes were incubated with the antibodies overnight at 4 °C. After several washings, membranes were incubated for 1 h at room temperature with the corresponding secondary antibody. Signals were detected using the western blot lightning plus-ECL reagent (PerkinElmer) and an Intas ChemoCam Imager 3.2 (Intas). Images were quantified using the Fiji software package.

### Indirect Immunofluorescence analysis
A549 cells were grown on glass coverslips, transfected or infected as described in the Results section and 48 h later, cells were fixed with 4% paraformaldehyde (Applichem GmbH, Darmstadt, Germany) for 20 min. Cells were permeabilized with 0.1% (v/v) Triton X-100 in PBS, incubated with primary antibodies in 1% BSA/PBS, washed and stained with the corresponding fluorescent secondary antibody. Coverslips were mounted onto glass slides with DAPI-Fluoromount-G (Southern BioTech) and images were acquired using a Zeiss confocal AiryScan 2 LSM900 or a Leica SP8 confocal microscope. Images were analyzed and quantified using the Fiji software package.

### Lipid peroxidation detection using coumarin hydrazide staining
A549 cells were infected with ZIKV (MOI = 3) and 5 h later, 200 µM 7-(diethylamino)-coumarin-3-carbohydrazide (Sigma Aldrich) was added to the cells. After 3 h, cells were washed 3 times with PBS prior to imaging using a Nikon Eclipse Ti microscope. Fluorescence intensity of 40 cells/experiment was determined using the Fiji software package. As positive and negative controls, cells were treated with Liproxstatin-1 (10 µM) or RSL-3 (1 µM) for 7 h prior to imaging.

### ROS detection using DCFDA staining
A549 cells were infected with ZIKV (MOI = 3) and 24 h later, 2′,7′-Dichlorofluorescein diacetate (DCFDA) (Sigma-Aldrich) at a final concentration of 25 µM was added to the cells. After 30 min, cells were washed 3 times with PBS prior to imaging using a Nikon Eclipse Ti microscope. Fluorescence intensity of 40 cells/experiment was determined using the Fiji software package.

### Immunoprecipitation of m⁶A-containing RNA
Total cellular RNA was isolated using the NucleoSpin RNA Plus extraction kit (Macherey-Nagel) as recommended by the manufacturer. Purified RNA was pre-cleared by incubation with Protein A beads for 1 h at 4 °C to remove non-specific binders. Remaining supernatants were incubated with 5 µg of m⁶A-specific antibody (synaptic system) or control IgG (Cell signaling). After 2 h incubation at 4 °C in IP buffer (10 mM Tris-HCl [pH 7.4], 150 mM NaCl, 0.1% NP-40 and 10U RNase inhibitor), Samples were incubated with protein A agarose beads at 4 °C for 2 h. After 3 washes with IP buffer, bound RNAs were eluted with 2×100 uL of elution buffer (6.7 mM m⁶A, 40U RNase

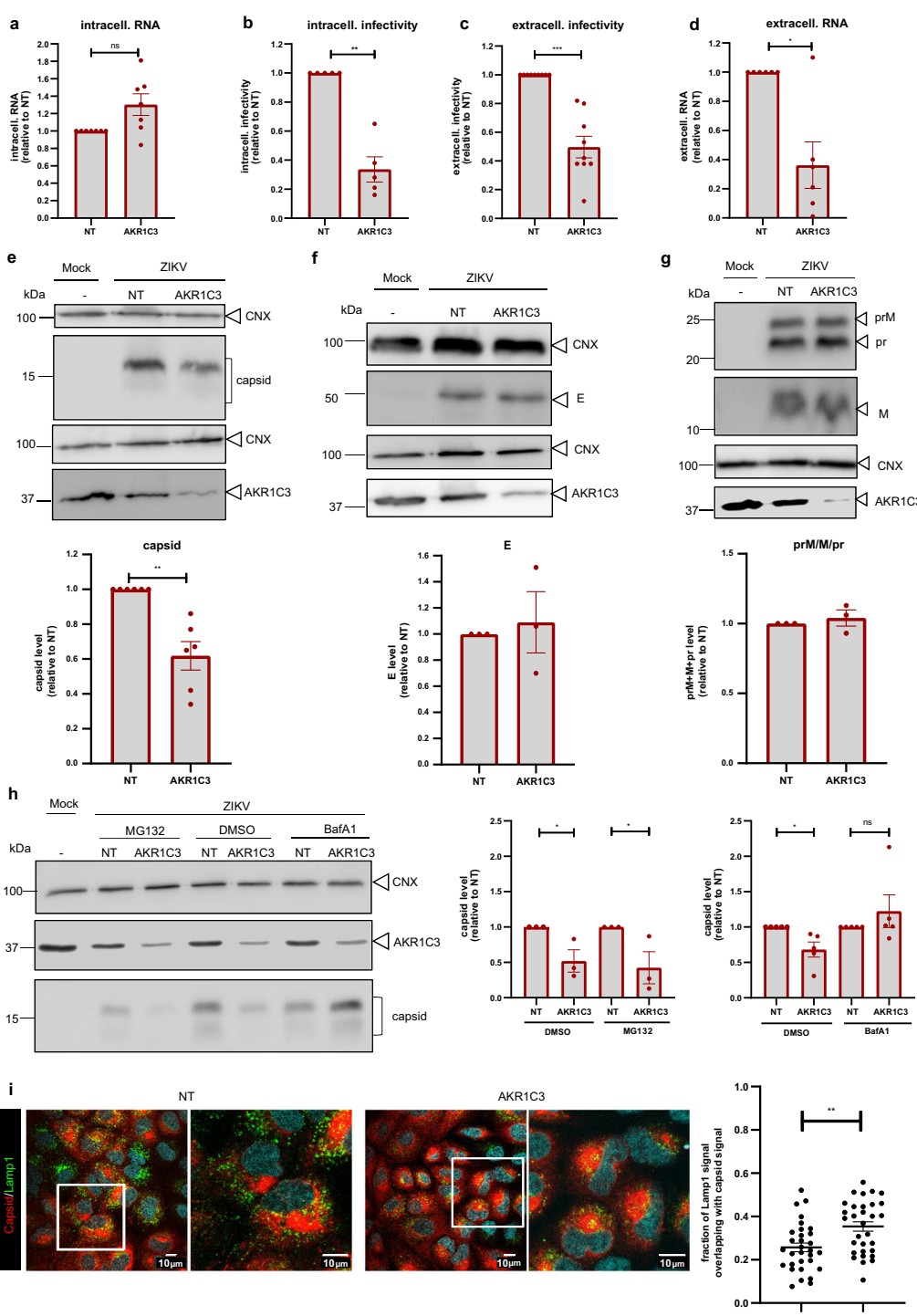

**Fig. 7 | AKR1C3 regulates ZIKV assembly by preventing lysosomal degradation of capsid.** A549 cells were transfected with AKR1C3-targeting or NT siRNAs for 48 h, followed by ZIKV infection (MOI = 3). Samples were harvested 24 h post infection. **a** Intracellular viral RNA (mean ± SEM, $n = 7$, $p = 0.0502$). **b** ZIKV intracellular infectivity (mean ± SEM, n = 5, $p < 0.0001$). **c** ZIKV extracellular infectivity (mean ± SEM, n = 9, $p < 0.0001$). **d** Extracellular viral RNA (mean ± SEM, $n = 6$, $p < 0.0001$). **e** Cell lysates analyzed by western blot (top) and quantified capsid signal normalized to the CNX loading control (bottom) (mean ± SEM, $n = 6$, $p = 0.0056$). **f** Cells lysates analyzed as in **e**, but for E protein (mean ± SEM, $n = 3$, $p = 0.7379$). **g** Same as in **e** but for prM, M and pr (mean ± SEM, $n = 5$, $p = 0.5653$). For panels **a**–**g**, statistical significance was assessed by a one-sample t-test (two-tailed). **h** A549 cells were transfected with AKR1C3-targeting or NT siRNAs for 48 h, followed by ZIKV infection (MOI = 3). Cells were treated with DMSO or Bafilomycin A1 (Baf A1; 50 nM) or MG132 (1 μM) 8 h post infection and samples were harvested 24 h

post infection. Cell lysates were analyzed by western blot (left) and capsid signal was quantified and normalized to the CNX loading control (right). Values were normalized to those obtained with NT control cells (for MG132: mean ± SEM, n = 3, one-way ANOVA with Holm-Sidak's multiple comparisons test (NT vs AKR1C3), $p = 0.0366$ (DMSO), $p = 0.0397$(MG132), for BafA1: mean ± SEM, $n = 5$, one-way ANOVA with Dunn's multiple comparisons test (NT. vs. AKR1C3), $p = 0.0414$ (DMSO), $p = 0.4834$ (BafA1). **i** Immunofluorescence of A549 cells transfected with AKR1C3-targeting or NT siRNAs for 48 h followed by ZIKV infection (MOI = 3) for 24 h and stained for ZIKV capsid (red) and Lamp-1 (green). Scale bars represent 10 μm for both panels and zooms. Degree of colocalization between Lamp-1 and capsid as determined with Manders' correlation coefficient of Lamp-1 signal overlapping with capsid signal is shown in the right panel. $n = 4$ with 5–10 cells/experiment. Unpaired t-test (two tailed) $p = 0.0019$. Source data are provided as a Source Data file.

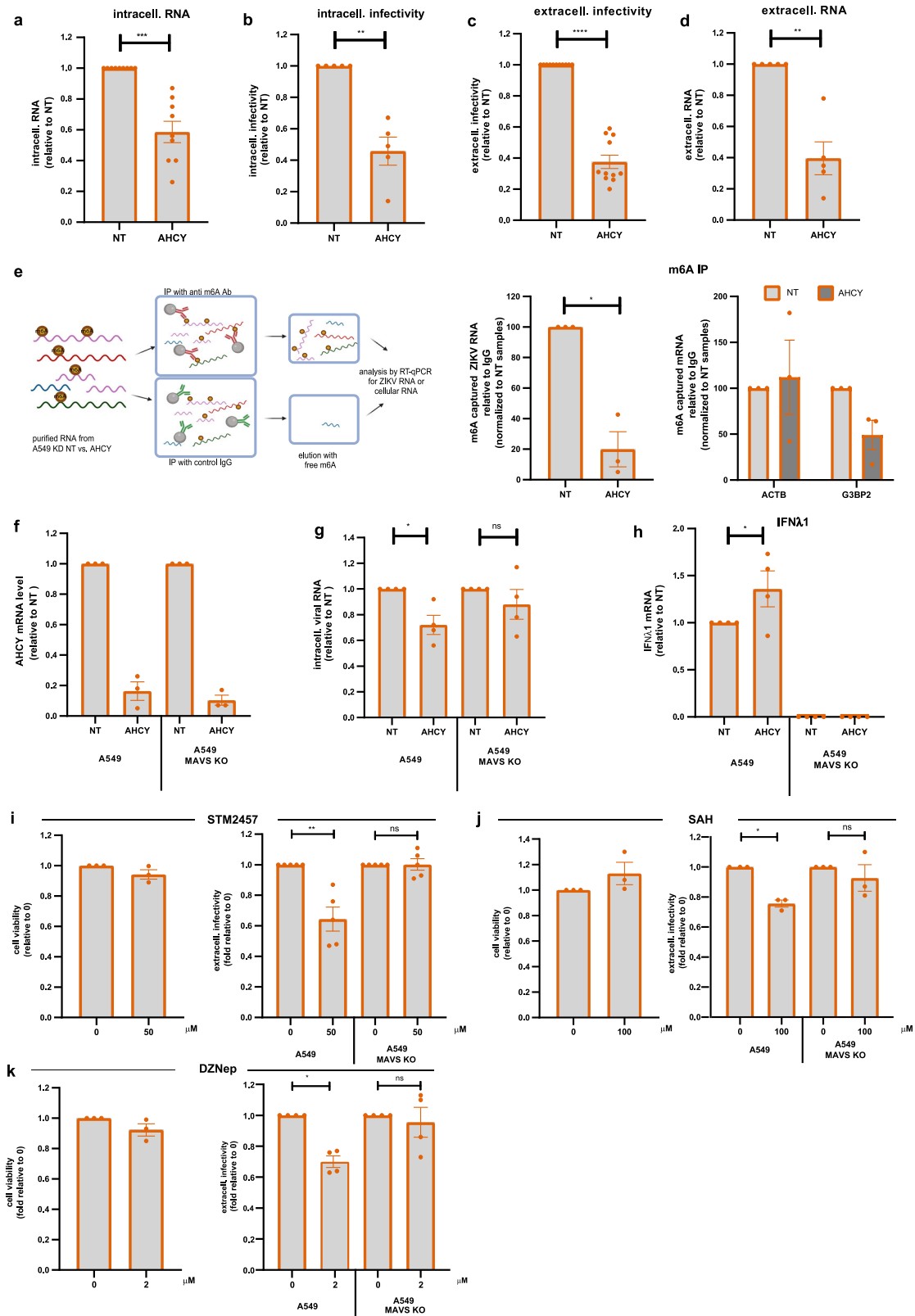

inhibitor) and precipitated with EtOH. Exogenous HCV RNA was added as calibrator for normalization and captured RNA was quantified by RT-qPCR as described above.

## Determination of apoE level by ELISA

apoE level in supernatants was determined using human apoE ELISA (Cell biolabs) according to manufacturer's instruction.

## Liproxstatin-1 treatment of cells with pIRO-Z system

Huh7-Lunet/T7 cells stably expressing the RNA polymerase of bacteriophage T7 were transfected with a pIRO-Z DNA construct encoding the ZIKV replicase proteins NS1 to NS5 as described previously[71]. Briefly, 500 ng of plasmid was transfected in 24well plate with Mirus TransIT-LT1 (ratio 1:3). After 4 h, medium was replaced with medium containing DMSO or Liproxstatin-1 (10 μM) (Sellekchem). Cells were

**Fig. 8 | AHCY regulates ZIKV replication by promoting methylation of viral RNA to escape intracellular sensing.** A549 or A549-MAVS KO cells were transfected with AHCY-targeting or NT siRNAs for 48 h, followed by ZIKV infection (MOI = 3). Samples were harvested 24 h post infection. **a** Intracellular viral RNA (mean ± SEM, *n* = 9, *p* < 0.0001). **b** ZIKV intracellular infectivity (mean ± SEM, *n* = 5, *p* < 0.0001). **c** ZIKV extracellular infectivity (mean ± SEM, n = 11, *p* < 0.0001). **d** Extracellular ZIKV RNA (mean ± SEM, *n* = 5, *p* < 0.0001). **e** M⁶A containing viral RNA, or ACTB or G3BP2 cellular mRNAs contained in lysates of siRNA transfected A549 cells were captured using m⁶A-specific immunoprecipitation (IP) and quantified by RT-qPCR (left panel, created with Biorender.com). Mean ± SEM, *n* = 3, *p* = 0.0201. For panels **a–e**, statistical significance was assessed with one-sample t-test (two-tailed). **f** AHCY mRNA level (mean ± SEM, n = 3). **g** ZIKV extracellular infectivity. Values were normalized to those obtained with NT control cells (mean ± SEM, *n* = 4, Kruskal-Wallis test with a Dunn's multiple comparisons test, *p* = 0.0223 (A549), *p* > 0.9999 (A549-MAVS KO)). **h** IFNλ1 mRNA amounts (mean ±

SEM, *n* = 4, one-way ANOVA with Sidak's multiple comparisons test (NT vs. AHCY), *p* = 0.0205). **i** A549 cells were treated with no or 50 μM STM2457 and cell viability was assessed 23 h post treatment (left) (mean ± SEM, *n* = 3). A549 or A549-MAVS KO were infected with ZIKV (MOI = 3) and treated with no or 50 μM STM2457 1 h post infection. Samples were harvested 24 h post infection. ZIKV extracellular infectivity (right). Values were normalized to those obtained with no treatment cells (mean ± SEM; *n* = 5 (right), Kruskal-Wallis test with a Dunn's multiple comparisons test, *p* = 0.0069(A549), p > 0.9999 (A549-MAVS KO)). **j** Same as **i** but treatment of cells with no or 100 μM SAH (mean ± SEM, *n* = 3, Kruskal-Wallis test with a Dunn's multiple comparisons, *p* = 0.0364(A549), p > 0.9999 (A549-MAVS KO)). **k** Same as **i** but treatment of cells with no or 2 μM DZNep (mean ± SEM, *n* = 3–4 independent experiments; Kruskal-Wallis test with a Dunn's multiple comparisons; *p* = 0.0347(A549), p > 0.9999 (A549-MAVS KO)). Source data are provided as a Source Data file.

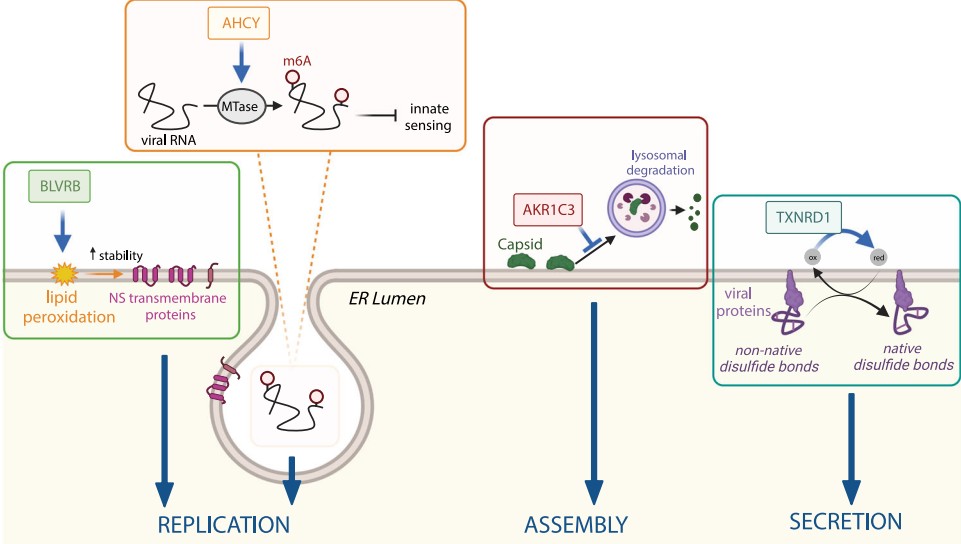

**Fig. 9 | Summary of the possible roles of host cell proteins enriched at ZIKV remodeled ER membranes in the viral life cycle.** See text for more details. Created with Biorender.com.

lysed 18 h post transfection and analyzed by western blot. For the experiment with MG132, MG312 (0.5 μM) (Sigma-Aldrich) was added at the same time as Liproxstatin-1.

## Viral replication assay
10 μg plasmid DNA encoding for the ZIKV subgenomic reporter replicon system (pFK-sgZIKV-R2A)[30] were linearized using the XhoI restriction enzyme (New England Biolabs) and subsequently purified using the NucleoSpin Gel and PCR clean-up kit (Macherey-Nagel). In vitro transcription of linearized DNA was performed as described previously[30]. Briefly, it was carried out with 10 μg of linearized plasmid DNA with 20 μL 5× RRL buffer (400 mM HEPES (pH 7.5), 60 mM MgCl2, 10 mM spermidine and 200 mM DTT), NTP-Mix (3.125 mM ATP, CTP and UTP and 1.56 mM GTP), 1 U/μL RNasin (Promega, Madison, WI, USA), 2 U/μL T7 RNA polymerase (New England Biolabs) and 1 mM anti-reverse cap analogue (ARCA; 3′-O-Me- m7G(5′)ppp(5′)G; New England Biolabs) for 2.5 h at 37 °C before addition of 1U/ μL T7 RNA polymerase for an additional incubation for 2.5 h at 37 °C. DNA was digested with DNaseI for one hour and RNA was purified by acidic phenol-chloroform extraction and isopropanol precipitation. For viral RNA transfection, sub-confluent Huh7-Lunet/T7 cells stably expressing firefly luciferase (Huh7-Lunet/T7-FLuc) were trypsinized and collected in DMEM complete. Cells were washed once with sterile PBS and subsequently resuspended in cytomix (120 mM KCl, 150 μM CaCl2, 10 mM potassium phosphate buffer (pH 7.6), 25 mM HEPES, 2 mM

EGTA, 5 mM MgCl₂, freshly supplemented with 5 mM glutathione and 2 mM ATP) at a density of 1 × 10⁷ cells/ml. Cell suspension (200 μl) was mixed with 5 μg in vitro transcribed viral RNA, transferred to an electroporation cuvette (0.2 cm gap width; Bio-Rad Laboratories) and pulsed once at 166 V and 975 μF using the Gene Pulser II (Bio-Rad Laboratories). Electroporated cells were resuspended in pre-warmed DMEM complete and seeded into 96-well plates at a density of 2.5 × 10⁴ cells/well in technical triplicates. Two hours post transfection, medium was replaced with DMEM complete containing Liproxstatin-1 (5–10 μM) or DMSO as negative control. Four and 24 h post-transfection, renilla and firefly luciferase activities as indicators of viral RNA content and cell viability, respectively, were determined on a Mithras LB940 plate luminometer (Berthold Technologies) using the Dual-Glo Luciferase Assay System (Promega) as described by the manufacturer. Data were analyzed using Microsoft Excel. To correct for potential cytotoxic effects and differences in luciferase decay rates between the time-points and biological replicates, renilla activity was normalized to firefly activity and mock-treated cells, respectively. To further account for variability in initial cellular RNA content, values of the 24 h time-point were normalized to those of the 4 h time-point.

## Cell viability assay
Cell viability was determined using CellTiter-Glo Luminescent Cell Viability Assay (Promega) following manufacturer's instruction.

## Statistical analysis

Statistical analyses excluding mass spectrometry data, were performed with the GraphPad Prism 8.0 software package (LaJolla, CA, USA). For all panels, each dot in the graphs corresponds to the value of an individual experiment. To assess statistical significance, statistical tests were used as indicated in the figure legends. Data sets were considered significantly different if the P value was less than 0.05. For each experiment, details and sample sizes are listed in the figure legends. Statistical significances are depicted by asterisks in the figures as follows: (*) for $p < 0.05$, (**) for $p < 0.01$, (***) for $p < 0.001$ and (****) for $p < 0.0001$.

## Reporting summary

Further information on research design is available in the Nature Portfolio Reporting Summary linked to this article.

## Data availability

Source data are provided with this paper. The mass spectrometry proteomics data generated in this study have been deposited via the ProteomeXchange Consortium in the PRIDE partner repository with the dataset identifier PXD043372. Source data are provided with this paper.

## Code availability

The scripts used for analysis are deposited in Zenodo and accessible via the following accession numbers: https://doi.org/10.5281/zenodo.8381245, https://doi.org/10.5281/zenodo.7746897; https://doi.org/10.5281/zenodo.7752068.

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

## Acknowledgements

We thank Ulrike Herian for excellent technical assistance, Alessia Ruggieri, Yannick Stahl and Sarah Goellner for precious help, advices and assistance and Antonio Piras for MS analysis. We are grateful to Didier Trono (EPFL, Lausanne, Switzerland) for providing the lentivirus packaging constructs. We thank Vibor Laketa and the Infectious Diseases Imaging Platform (IDIP) for facility use and expert support. We are grateful to all members of the Molecular Virology unit for continuous stimulating discussions. R.B. is supported by grants from the Deutsche Forschungsgemeinschaft (DFG, German Research Foundation; project number 240245660–SFB1129, and project number 499982526-BA 1505/11-1) and the Helmholtz Association's Initiative and Networking Fund KA1-Co-02 "COVIPA" (Virological and immunological determinants of COVID-19 pathogenesis – lessons to get prepared for future pandemics). S.D. was supported by a Humboldt Research Fellowship for Postdoctoral Researchers. T.P.D. is supported by grants from the DFG (TRR186, SPP2306) and the European Commission (742039). A.P. was supported by the European Research Council (ERC-CoG, Grant no. 817798), Bavarian State Ministry of Science and Arts (Bavarian Research Network FOR-COVID), the German Research Foundation (DFG) (TRR237 (A07), TRR179 (TP11)) and the Helmholtz Association's Initiative and Networking Fund KA1-Co-02 "COVIPA".

## Author contributions

Conceptualization and methodology, S.D., R.B.; Formal analysis, S.D., A.S., Investigation, S.D., A.S., U.B., A.R., S.J, P.K.; Writing-Original Draft, S.D., R.B.; Supervision, S.D., R.B., A.P., T.D.; Funding Acquisition, R.B., A.P., T.D.

## Funding

## Competing interests

The authors declare no competing interest.
