## [Peer Review File · Nature Communications]

Zika virus remodelled ER membranes contain proviral factors involved in redox and methylation pathwaysREVIEWER COMMENTS

Reviewer #1 (Remarks to the Author):

In this manuscript, the authors prepared ER membranes from Zika virus (ZIKV)-infected cells and conducted proteomic analysis to determine the host proteins that are associated and enriched in these membranes. The analysis revealed at least 46 cellular proteins that were enriched in ZIKV-infected ER membranes. Interestingly, gene ontology (GO) analysis showed strong enrichment of proteins involved in oxidoreductase reactions and in RNA methylation pathway. The authors then conducted further studies to determine the role of some of these proteins in viral genome replication and infectious progeny production. Focusing on three proteins involved in cellular redox homeostasis/metabolic interconversion pathways [thioredoxin reductase 1 (TXNRD1), biliverdin reductase B (BLVRB), aldo-keto reductase family 1 member C3 (AKR1C3)] and one protein in RNA methylation pathway [adenosylhomocysteinase (AHCY)], the authors have demonstrated that depletion of these cellular proteins by siRNA-mediated knockdown affected either infection progeny production, viral RNA replication, or methylation status of the viral genome. The authors conclude that these four proteins support different steps in the viral replication process including stability, proper folding and secretion of viral proteins, and methylation of viral RNA that likely plays a role in immune evasion.

Overall, the studies presented here are carefully conceived with appropriate controls. The interpretation of the results is appropriate and the conclusions are consistent with the data. However, the effects of the knockdown of the proteins on viral replication steps do not appear to be robust or highly significant. The studies and the results in the manuscript are new, interesting and advance our understanding of how ZIKV recruits various cellular factors to the ER membrane compartment to aid in its genome replication and infectious progeny production. Although ZIKV is known to induce oxidative stress resulting in redox imbalance in the infected cell with some protein factors in the redox pathway acting either as proviral or antiviral, the current study has identified several players in the pathway that associate with ER membranes and positively impact steps in viral replication cycle. The authors have gone on to understand the role of these proteins in the virus life cycle. The manuscript is clearly written. My specific comments are as follows:

1. pp. 4, line 95: The authors claim that knockdown of TXNRD1 "reduced the production of extracellular infectious virus"; however, the reduction is only less than 40%. Statistical analysis of the results confirm that the reduction is also not significant. Additional data presented in Fig. 4 also show that various steps of virus life cycle are only marginally inhibited by the knockdown. I just wonder how significant a role TXNRD1 plays in this virus life cycle. The authors need to explain why they think this cellular factor is important in ZIKV replication.

2. If LPO alters membrane fluidity and permeability, viral proteins anchored to the membrane may likely be more unstable. In that scenario, how does LPO enhance ZIKV RNA replication?

3. pp. 6, line 148: Figure 4H – Should this be Figure 4I?

4. The effects of knockdown of BLVRD or AHCY on intracellular RNA levels and extracellular progeny production (Fig. 3C-D) are also not significant. Why did the authors choose to focus on these two factors?

5. What are the horizontal dotted lines in the figures?

Reviewer #2 (Remarks to the Author):

This manuscript seeks to characterize the proteins present at the ER during Zika virus (ZIKV) infection to identify novel regulators of viral infection, with a particular emphasis on host factors that could be acting at the sites of viral replication. To identify ZIKV-relocalized ER proteins, first, ER membranes were isolated by using a combination of membrane enrichment via density gradient centrifugation with the fractions containing viral markers pooled. These pooled fractions were then further purified by affinity capture for a tagged-ER protein. Then, the eluates of this capture were analyzed by mass spectrometry. This is an innovative approach, and the controls validate that the approach is working as described, specifically that viral proteins, along with the tagged-ER protein and the (-) strand replicative intermediate RNA, are all present. Secondary siRNA screening of the 46 proteins for viability, entry and replication, and virus production led to 13 top candidates for host factors, of which 4 passed validation screening.

The next part of the manuscript sought to figure out how these 4 proteins were acting as host factors in ZIKV replication. Each figure started with a characterization of their role in the ZIKV life cycle by measuring intracellular RNA, intracellular infectivity, extracellular infectivity, and extracellular RNA following depletion of the factor by siRNA. Then, a description of the factor was provided in the text, along with some experiments that tried to show what the factor was doing in ZIKV infection.

The factors did not seem to be acting in the same pathways, indicating that ER-associated host factors can regulate viral infection through diverse mechanisms. However, while the screen was robust and exciting, the characterization of the four hits identified was more superficial, often without orthogonal validation, and as such the conclusions were primarily correlative. We suggest in-depth characterization of 1-2 of these factors with orthogonal approaches and additional controls on the other ones to increase the impact of the conclusions that can be made from the work.

Specific Comments:

1. The conclusions of Figure 4 on the role of TXNRD1 are largely correlative. It is clear that loss of TXNRD1 reduces secretion of ZIKV infectious particles and also the secretion of the NS1 protein. Does over-expression lead to the opposite result? Data presented in this figure also suggests that TXNRD1, which promote disulfide bond formation of ER-resident proteins, is required for proper folding of NS1. Experiments to link these concepts are required to know if the function of TXNRD1 in disulfide bond formation is the function required for virion secretion. Further, experiments that directly prove that TXNRD1 is altering disulfide bond formation in NS1 are important. As it stands, it could be that it alters a chaperone required for folding of NS1. An additional control examining the role of TXNRD1 in secretion of host protein(s) with and without disulfide bridges would be useful.
2. As the described roles of BLVRB are only indirectly linked to lipid peroxidation, establishing these links is important to make conclusions about this protein could be regulating ZIKV infection. Does BLVRB regulate the levels of ROS generated during ZIKV? Is this the link between BLVRB and lipid peroxidation? If you prevent ROS generation, can you alleviate the need for BLVRB during ZIKV? Is the enzymatic activity of BLVRB required for its function? How does this impact the protein levels of the viral transmembrane proteins (similar to Fig. 6C)?
3. Figure 6 implies that the role of BLVRB in ZIKV is to induce lipid peroxidation to promote the stability of viral transmembrane proteins; this is interesting as lipid peroxidation in hepatitis C virus has the opposite effect – inhibits replication. In Figure 6, liproxstatin was used to induce inhibit lipid peroxidation in the presence of BLVRB, resulting in decreased levels of viral transmembrane replicase proteins. A more robust measure of protein synthesis (e.g. pulse chase) should be used here – is it that the proteins are not made or that they are not stable? What is the effect of BLRVB KD on the stability of the viral proteins shown here?
4. In Figure 7, examining Capsid localization via microscopy could strengthen the conclusions drawn from this data, to determine if capsid relocalizes to the lysosomes in AKR1C3 KD cells.
5. In Figure 8, to link AHCY more concretely to the decrease in m6A on ZIKV, control experiments showing that loss of m6A on ZIKV via loss of the m6A-addition enzymes phenocopies the effects would be important. This could be done by siRNA to METTL3/METTL14 or by using the

commercially available METTL3 inhibitor. Further, are the levels of SAH increased by AHCY depletion, or if you supplement cells with SAH do you see the same effects (less virus in a MAVS-dependent fashion)? Alternatively, this point could also be addressed by determining if the enzymatic activity of AHCY is required. Are cellular transcripts affected by these changes?

6. In addition, the abstract reads as if the authors believe that AHCY is part of the m6A writer complex, which was clearly not the intent, as the conclusions in the results and discussion were carefully written. I might suggest a slight rewording to imply the indirect nature of the regulation. For example, perhaps something like "promoting methylation of ZIKV RNA by decreasing SAH levels to promote m6A activity by the known machinery".

7. Are the four factors identified unique to ZIKV, or do they also have roles in the related flavivirus, DENV?

8. Do the authors think that the subcellular relocalization of these factors is essential for their regulation of ZIKV? It would add to the discussion to include this topic.

Reviewer #3 (Remarks to the Author):

In the manuscript NCOMMS-23-26853 entitled "Zika virus remodelled ER membranes proviral factors involved in redox and methylation pathways" the authors identified proteins overrepresented or depleted in ER membranes isolated from cells infected with Zika virus vs uninfected cells. They found 54 such proteins, including certain redox and methylating enzymes. Four of these enzymes (TXNRD1, AKR1C3, BLVRB, and AHCY) were knocked-down and the effects in the context of zika virus infection studied using standard assays. Overall, the paper is well written, organized and the data seems to support the conclusions of the paper. The following are my specific comments.

1. The purification of ER membranes. Has this 2-step procedure (fractionation + IP) been published previously? If this or similar approach have been used, please cite.
2. The proteomics data.
 - a. How many replicates were used in the proteomics experiment? Were they technical or biological replicates?
 - b. In the Supplementary Table 1 some important information is missing, please add the following: number razor+unique peptides and LFQ values for individual samples.
 - c. Four comparisons of protein levels were performed: Mock calnexin vs Mock empty, Zika calnexin vs Zika empty, Zika calnexin vs Mock calnexin, Zika empty vs Mock empty and the result contained in Supplementary Table 1. In the main text, the authors stated that 46 protein were enriched and 8 depleted in the membranes isolated from zika infected vs non-infected cells. Could the authors elaborate how they arrived at these numbers, please? Which statistical tests were performed? Which were the thresholds? Were all four comparisons used to arrive at the final list of 54 differentially regulated proteins?
 - d. There is a vague information in the methods section about proteomics data processing, please include MaxQuant parameters used for the search and list all the data processing steps performed post MaxQuant.
 - e. Have any data imputation been done to LFQ values? If so, please explain how, in the methods section.
- d. The bait for affinity purification (CANX) should be highlighted in Fig2A. The authors should consider adding volcano plots for all 4 comparison as supplementary figures.

Point-by-point response:

In the following, we address each comment item-by-item. Our replies are marked in blue letters.

REVIEWER COMMENTS

Reviewer #1 (Remarks to the Author):

In this manuscript, the authors prepared ER membranes from Zika virus (ZIKV)-infected cells and conducted proteomic analysis to determine the host proteins that are associated and enriched in these membranes. The analysis revealed at least 46 cellular proteins that were enriched in ZIKV-infected ER membranes. Interestingly, gene ontology (GO) analysis showed strong enrichment of proteins involved in oxidoreductase reactions and in RNA methylation pathway. The authors then conducted further studies to determine the role of some of these proteins in viral genome replication and infectious progeny production. Focusing on three proteins involved in cellular redox homeostasis/metabolic interconversion pathways [thioredoxin reductase 1 (TXNRD1), biliverdin reductase B (BLVRB), aldo-keto reductase family 1 member C3 (AKR1C3)] and one protein in RNA methylation pathway [adenosylhomocysteinase (AHCY)], the authors have demonstrated that depletion of these cellular proteins by siRNA-mediated knockdown affected either infection progeny production, viral RNA replication, or methylation status of the viral genome. The authors conclude that these four proteins support different steps in the viral replication process including stability, proper folding and secretion of viral proteins, and methylation of viral RNA that likely plays a role in immune evasion.

Overall, the studies presented here are carefully conceived with appropriate controls. The interpretation of the results is appropriate and the conclusions are consistent with the data. However, the effects of the knockdown of the proteins on viral replication steps do not appear to be robust or highly significant. The studies and the results in the manuscript are new, interesting and advance our understanding of how ZIKV recruits various cellular factors to the ER membrane compartment to aid in its genome replication and infectious progeny production. Although ZIKV is known to induce oxidative stress resulting in redox imbalance in the infected cell with some protein factors in the redox pathway acting either as proviral or antiviral, the current study has identified several players in the pathway that associate with ER membranes and positively impact steps in viral replication cycle. The authors have gone on to understand the role of these proteins in the virus life cycle. The manuscript is clearly written.

Response: we thank Reviewer#1 for the encouraging appreciation of our work.

My specific comments are as follows:

1. pp. 4, line 95: The authors claim that knockdown of TXNRD1 “reduced the production of extracellular infectious virus”; however, the reduction is only less than 40%. Statistical analysis of the results confirm that the reduction is also not significant. Additional data presented in Fig. 4 also show that various steps of virus life cycle are only marginally inhibited by the knockdown. I just wonder how significant a role TXNRD1 plays in this virus life cycle. The authors need to explain why they think this cellular factor is important in ZIKV replication.

Response: We agree with Reviewer #1 that the effect of depleting TXNRD1 was relatively mild.

However, it was reproducible in all experiments and to make this clearer, we have modified the bar charts in **Figure 3** to show the individual measurements. Importantly, in **Figure 4** we show additional analysis demonstrating statistically significant reduction of extracellular infectivity and extracellular viral RNA upon depletion of TXNRD1. Interestingly, depletion of TXNRD1 also led to a two-fold reduction of NS1 secretion. Moreover, we now show that TXNRD1 depletion also impairs the life cycles of DENV and YFV (**Supplementary Figure 5**), indicating that TXNRD1 plays a more general role in flaviviruses, even though it appears to be a rather subtle one.

2. If LPO alters membrane fluidity and permeability, viral proteins anchored to the membrane may likely be more unstable. In that scenario, how does LPO enhance ZIKV RNA replication?

Response: We thank this Reviewer for the opportunity to discuss this point. It is generally difficult to predict the influence of LPO on biophysical membrane properties. Previous studies showed that LPO can either increase or decrease membrane fluidity, depending on composition and context. For example, it was shown that cholesterol modulates the effects of LPO. As we do not know the lipid composition of viral vesicle packets, we can only speculate on how LPO may enhance replication. Our data suggest that viral transmembrane proteins are less stable when LPO is inhibited, suggesting that LPO creates a membrane environment conducive to the insertion and/or stabilization of viral transmembrane proteins. It is also possible that viral transmembrane proteins are post-translationally modified by reactive lipid species (as generated by LPO) to enhance their stability and/or function (reviewed in (Viedma-Poyatos et al. 2021)). We modified the text to discuss these points.

3. pp. 6, line 148: Figure 4H – Should this be Figure 4I?

Response: we thank Reviewer#1 for pointing out this mistake. We corrected the text.

4. The effects of knockdown of BLVRD or AHCY on intracellular RNA levels and extracellular progeny production (Fig. 3C-D) are also not significant. Why did the authors choose to focus on these two factors?

Response: Similar to TXNRD1 (see point 1), we decided to focus on these two proteins because their knock-down led to reproducible effects in at least three independent experiments (**Figures 3C-D**). Moreover, these effects have been confirmed in subsequent validations revealing statistical significance (**Figures 5A and 8A**).

5. What are the horizontal dotted lines in the figures?

Response: the dotted lines are now defined in the figure legend. They are meant to serve as visual guidelines. This is now mentioned in the figure legend.

Reviewer #2 (Remarks to the Author):

This manuscript seeks to characterize the proteins present at the ER during Zika virus (ZIKV) infection to identify novel regulators of viral infection, with a particular emphasis on host factors that could be acting at the sites of viral replication. To identify ZIKV-relocalized ER proteins, first, ER membranes were isolated by using a combination of membrane enrichment via density gradient centrifugation with the fractions containing viral markers pooled. These pooled fractions were then further purified by affinity capture for a tagged-ER protein. Then, the eluates of this capture were analyzed by mass spectrometry. This is an innovative approach, and the controls validate that the approach is working as described, specifically that viral proteins, along with the tagged-ER protein and the (-) strand replicative intermediate RNA, are all present. Secondary siRNA screening of the 46 proteins for viability, entry and replication, and virus production led to 13 top candidates for host factors, of which 4 passed validation screening.

The next part of the manuscript sought to figure out how these 4 proteins were acting as host factors in ZIKV replication. Each figure started with a characterization of their role in the ZIKV life cycle by measuring intracellular RNA, intracellular infectivity, extracellular infectivity, and extracellular RNA following depletion of the factor by siRNA. Then, a description of the factor was provided in the text, along with some experiments that tried to show what the factor was doing in ZIKV infection.

The factors did not seem to be acting in the same pathways, indicating that ER-associated host factors can regulate viral infection through diverse mechanisms. However, while the screen was robust and exciting, the characterization of the four hits identified was more superficial, often without orthogonal validation, and as such the conclusions were primarily correlative. We suggest in-depth characterization of 1-2 of these factors with orthogonal approaches and additional controls on the other ones to increase the impact of the conclusions that can be made from the work.

Response: we thank Reviewer#2 for this general feedback. We now provide additional experiments to deepen the characterization of the factors according to the suggestions.

Specific Comments:

1. The conclusions of Figure 4 on the role of TXNRD1 are largely correlative. It is clear that loss of TXNRD1 reduces secretion of ZIKV infectious particles and also the secretion of the NS1 protein. Does over-expression lead to the opposite result? Data presented in this figure also suggests that TXNRD1, which promote disulfide bond formation of ER-resident proteins, is required for proper folding of NS1. Experiments to link these concepts are required to know if the function of TXNRD1 in disulfide bond formation is the function required for virion secretion. Further, experiments that directly prove that TXNRD1 is altering disulfide bond formation in NS1 are important. As it stands, it could be that it alters a chaperone required for folding of NS1. An additional control examining the role of TXNRD1 in secretion of host protein(s) with and without disulfide bridges would be useful.

Response: we performed additional experiments to further characterize the role of TXNRD1 in the ZIKV life cycle.

1) As suggested by the reviewer, we overexpressed TXNRD1. Overexpression of TXNRD1 did not influence ZIKV infectivity (Rebuttal Figure 1A-B). NS1 secretion seems to be impaired, although this effect was variable and below statistical significance (Rebuttal Figure 1C). We believe that this outcome is not unexpected. First, the endogenous expression level of TXNRD1 is unlikely to be a limiting factor for the secretion of viral components. Second, reductive and oxidative activities in the ER must be delicately balanced to achieve proper disulfide bond formation. Thus, increasing the reductive component (i.e., TXNRD1) may well perturb this balance, thus potentially worsening NS1 maturation.

Rebuttal Figure 1. Overexpression of TXNRD1. A549 cells were transduced with a lentiviral vector encoding TXNRD1. Two days after, cells were infected with ZIKV. Samples were harvested 24h post infection. (A) Titers of intracellular infectious ZIKV as assessed by plaque assay (mean \pm SEM of 4 independent experiments). (B) Titers

of extracellular infectious virus as assessed by plaque assay (mean \pm SEM of 4 independent experiments). (C) Lysates and supernatants of transduced and infected A549 cells analyzed by western blot to detect intra- (iNS1) and extracellular NS1 (eNS1), respectively (left panel). Signal intensities from 4 independent experiments were quantified and values were normalized to those obtained with non-transduced cells (right panel). Data represent the mean \pm SEM.

2) We agree with the Reviewer that the effect of TXNRD1 on oxidative NS1 folding is indirect. TXNRD1 is not involved in disulfide bond formation per se (which is catalyzed by the sulfhydryl oxidase ERO1 in combination with PDI family members). Instead, the oxidoreductase TXNRD1 provides the reducing equivalents (derived from NADPH) that are needed to reduce non-native (wrong) disulfide bonds, which are generated by the ERO1/PDI machinery (see Rebuttal Figure 2). Thus, TXNRD1 activity allows for the “correction” of wrongly formed disulfide bonds. Without such “correction”, the maturation of secretory proteins is less efficient, as there is more deficient (non-native) product. For example, this has been shown to be critical for folding and secretion of the LDL receptor (Poet et al. 2017).

Rebuttal Figure 2. Figure from (Poet et al. 2017)

Unfortunately, the reorganization of disulfide bonds (i.e., the replacement of a non-native disulfide by a native disulfide) is very difficult to demonstrate experimentally, especially if there is no change in the total number of disulfide bonds. We performed additional experiments in which we “froze” the thiol-disulfide status with an alkylating agent (NEM) prior to lysis, to conserve disulfide bonds during lysis. Interestingly, upon TXNRD1 depletion we could then see accumulation of high molecular weight NS1 aggregates under non-reducing condition, suggesting that NS1 is indeed forming non-native (inter-subunit) disulfide bonds, either with itself or with other ER proteins. We added these data to **Figure 4I** and modified the text accordingly. We also modified the text to better explain the role of TXNRD1.

3) In addition, we tested the effect of TXNRD1 depletion on two cellular proteins, one with and one without disulfide bonds, i.e., apolipoprotein E (apoE) and alpha-1-antitrypsin (A1AT), respectively. We found that TXNRD1 depletion decreased apoE but not A1AT secretion, as determined by ELISA and immunoblotting. These findings are in full agreement with a previous publication (Poet et al. 2017), showing that TXNRD1 only plays a role in the maturation of secretory proteins having disulfide bonds. We added these data to **Figures 4F-G** and modified the text accordingly.

2. As the described roles of BLVRB are only indirectly linked to lipid peroxidation, establishing these links is important to make conclusions about this protein could be regulating ZIKV infection. Does BLVRB regulate the levels of ROS generated during ZIKV? Is this the link between BLVRB and lipid peroxidation? If you prevent ROS generation, can you alleviate the need for BLVRB during ZIKV? Is the enzymatic activity of BLVRB required for its function? How does this impact the protein levels of the viral transmembrane proteins (similar to Fig. 6C)?

Response: In our manuscript we linked BLVRB to LPO by showing that RSL3, a chemical inducer of LPO, can compensate for BLVRB depletion (**Figure 5I**). This indicated to us that BLVRB supports ZIKV infectivity by acting as an inducer or enhancer of LPO. We now support our conclusions with additional experiments, as suggested by the reviewer.

1) We asked if BLVRB depletion influences ZIKV-induced ROS generation. To this end, we stained cells with DCFHDA, a fluorescent dye activated by ROS. We observed increased DCFHDA oxidation upon ZIKV infection, and this increase was blunted in BLVRB depleted cells. This finding supports the notion that BLVRB contributes to ROS formation in the context of ZIKV infection. The increase in ROS matches the increase in LPO, which is known to be initiated by ROS. We added these data to **Supplementary Figure 2B** and modified the text accordingly.

2) We treated cells with the antioxidant N-acetylcysteine (NAC) to mitigate endogenous ROS generation. Indeed, NAC treatment of infected cells lowered intracellular ZIKV RNA levels and decreased extracellular infectivity. Interestingly, NAC did not have an effect in cells depleted of BLVRB, suggesting that BLVRB deficiency impairs the ZIKV life cycle by preventing ZIKV-induced ROS generation. We added the data to **Figure 5J** and modified the text accordingly.

3) To assess the role of BLVRB enzymatic activity, we tested two previously established small molecule inhibitors, Phloxine B and lumichrome. We found that both inhibitors lower intracellular ZIKV RNA levels as well as extracellular infectivity, suggesting that BLVRB enzymatic activity supports the ZIKV life cycle. We added these data to **Supplementary Figure 3A** and modified the text accordingly.

4) We used the pIRO-Z system to evaluate the impact of BLVRB depletion on the stability of viral transmembrane proteins (see below). This system is more suitable to address the stability of proteins as it allows characterization of protein levels independently of replication and production of infectious particles, which are modulated by BLVRB KD. We provided more details in point 3.

3. Figure 6 implies that the role of BLVRB in ZIKV is to induce lipid peroxidation to promote the stability of viral transmembrane proteins; this is interesting as lipid peroxidation in hepatitis C virus has the opposite effect – inhibits replication. In Figure 6, liproxstatin was used to induce inhibit lipid peroxidation in the presence of BLVRB, resulting in decreased levels of viral transmembrane replicase proteins. A more robust measure of protein synthesis (e.g. pulse chase) should be used here – is it that the proteins are not made or that they are not stable? What is the effect of BLRVB KD on the stability of the viral proteins shown here?

Response: we thank the Reviewer for this comment. As viral proteins are expressed as a polyprotein and as the levels of NS1 and NS5, the first and the last protein of the polyprotein expressed in pIRO-Z system, are not impaired by Liproxstatin-1, we can exclude the possibility that there is an issue with protein synthesis. We added an immunoblot to **Supplementary Figure 2C**, to show that there is no defect in the processing of NS4A and NS4B. In addition, we treated transfected cells with the combination of Liproxstatin-1 and the proteasome inhibitor MG132. We observed that MG132 rescued Liproxstatin-1 induced degradation of NS4A and NS4B, while there was no change in NS3 and NS5, showing that Liproxstatin-1 treatment only results in the degradation of viral proteins having transmembrane domains. We added these data to **Supplementary Figure 3D** and changed the text accordingly.

We also tested the effect of BLVRB depletion on the stability of viral proteins using the pIRO-Z system. Interestingly, BLVRB depletion impaired the stability of NS2B, NS4A and NS4B, mirroring the effects

observed with Liproxstatin-1 treatment. We added these data to **Figure 6D** and modified the text accordingly.

4. In Figure 7, examining Capsid localization via microscopy could strengthen the conclusions drawn from this data, to determine if capsid relocates to the lysosomes in AKR1C3 KD cells.

Response: We examined capsid and Lamp-1 localization in AKR1C3 WT and KD cells. Interestingly, we could not see a major change in the subcellular distribution of the capsid signal but observed more overlap between the Lamp1 and capsid signals, as assessed by Mander's coefficient. This may reflect closer proximity between the capsid and the lysosome, in line with a possible lysosomal degradation of the capsid upon depletion of AKR1C3. We added these data to **Figure 7 (panel I)** and modified the text accordingly.

5. In Figure 8, to link AHCY more concretely to the decrease in m6A on ZIKV, control experiments showing that loss of m6A on ZIKV via loss of the m6A-addition enzymes phenocopies the effects would be important. This could be done by siRNA to METTL3/METTL14 or by using the commercially available METTL3 inhibitor. Further, are the levels of SAH increased by AHCY depletion, or if you supplement cells with SAH do you see the same effects (less virus in a MAVS-dependent fashion)? Alternatively, this point could also be addressed by determining if the enzymatic activity of AHCY is required. Are cellular transcripts affected by these changes?

Response: we thank the Reviewer for raising these points. We performed the following experiments:

1) We tested a METTL3 inhibitor, STM2457. We observed a decrease of ZIKV RNA level and production of infectious particles, in a MAVS dependent fashion, suggesting that inhibition of METTL3 activity can phenocopy AHCY KD. We added the data in **Figure 8I** and **Supplementary Figure 4B** and modified the text accordingly.

2) We tested SAH supplementation of cells. We could observe a slight but significant decrease of ZIKV RNA level and production of infectious particles, in a MAVS dependent fashion, suggesting that regulation of SAH might regulate ZIKV life cycle. We added the data in **Figure 8J** and **Supplementary Figure 4C** and modified the text accordingly.

We also tested an inhibitor of AHCY, DZNep, and again observed a significant decrease of ZIKV RNA level and production of infectious particles in a MAVS dependent fashion, suggesting that enzymatic function of AHCY is involved in ZIKV life cycle. We added the data in **Figure 8K** and **Supplementary Figure 4D** and modified the text accordingly.

3) We checked the m6A level of two cellular transcripts. We chose two transcripts which were shown to be m6A-modified but without modulation of this modification by ZIKV infection according to (Gokhale et al. 2020). To this end, we chose ACTB (Actin Beta) and G3BP2 (Ras-GTPase Activating Protein SH3 Domain-Binding Protein 2). We did not observe any impairment of ACTB m6A modification while we observed a decrease of G3BP2 m6A level, suggesting that AHCY might also regulate directly or indirectly m6A level of some cellular transcripts. We added these data in **Figure 8E** and modified the text accordingly.

6. In addition, the abstract reads as if the authors believe that AHCY is part of the m6A writer complex, which was clearly not the intent, as the conclusions in the results and discussion were carefully written. I might suggest a slight rewording to imply the indirect nature of the regulation. For example, perhaps something like "promoting methylation of ZIKV RNA by decreasing SAH levels to promote m6A activity by the known machinery".

Response: we thank the Reviewer for this valuable suggestion. We edited the abstract to clarify this point.

7. Are the four factors identified unique to ZIKV, or do they also have roles in the related flavivirus, DENV?

Response: We tested the effect of KD of TXNRD1, AKR1C3, BLVRB, and AHCY on DENV and YFV replication and virus production as determined by plaque assay. We observed reduced virus titers after KD of all factors for DENV and after KD of TXNRD1, AKR1C3, and BLVRB for YFV. We added the data in **Supplemental Figure 5** and modified the text accordingly.

8. Do the authors think that the subcellular relocalization of these factors is essential for their regulation of ZIKV? It would add to the discussion to include this topic.

Response: We now discuss these points in the revised manuscript (lines 373-382).

Reviewer #3 (Remarks to the Author):

In the manuscript NCOMMS-23-26853 entitled “Zika virus remodelled ER membranes proviral factors involved in redox and methylation pathways” the authors identified proteins overrepresented or depleted in ER membranes isolated from cells infected with Zika virus vs uninfected cells. They found 54 such proteins, including certain redox and methylating enzymes. Four of these enzymes (TXNRD1, AKR1C3, BLVRB, and AHCY) were knocked-down and the effects in the context of zika virus infection studied using standard assays. Overall, the paper is well written, organized and the data seems to support the conclusions of the paper.

Response: we thank this Reviewer for the positive comment.

The following are my specific comments.

1. The purification of ER membranes. Has this 2-step procedure (fractionation + IP) been published previously? If this or similar approach have been used, please cite.

Response: this protocol was established by us for this project, adapted from a former study of our lab (Paul et al. 2013). In this former study the procedure was used for the purification of hepatitis C virus induced double membrane vesicles. We modified the text to indicate this point.

2. The proteomics data

a. How many replicates were used in the proteomics experiment? Were they technical or biological replicates?

Response: we made the analysis on 6 independent biological replicates. For each batch independent ZIKV infection and affinity purification were performed.

b. In the Supplementary Table 1 some important information is missing, please add the following: number razor+unique peptides and LFQ values for individual samples.

Response: For our statistical analysis we were using precursor MS1 intensities (evidence.txt table) instead of MaxQuant LFQ intensities.

We were also using an alternative protein group definition: each protein group should have at least 2 peptides unique to that protein group (MaxQuant requires at least 1) or at least 25% of peptides of that protein group should be unique (protein groups candidates that share most of their peptides and don't have enough unique ones are merged). Unlike MaxQuant protein group definition, it is more persistent in assigning isoforms of the same gene to the same protein group and results in ~5% less protein groups, thus avoiding the generation of isoform-specific protein groups that contain too little specific peptides to conclude that such isoforms have distinct behavior upon CNX-binding or ZIKV infection.

For statistical analysis, only the intensities of the protein group-unique peptides were used (which are effectively unique+razor peptides of MaxQuant protein groups). In the revised manuscript, we have expanded the methods section to include these details. We have now also added the number of total and protein group-unique modified peptide sequences to our report table ("npepmods" and "npemods_unique" columns, respectively). We did not include LFQ intensities in the report to avoid confusion, since they are calculated for MaxQuant protein groups and were not used in the statistical analysis. However, they are still accessible as proteinGroups.txt via our PRIDE submission (PXD043372; will be released upon acceptance on our manuscript).

c. Four comparisons of protein levels were performed: Mock calnexin vs Mock empty, Zika calnexin vs Zika empty, Zika calnexin vs Mock calnexin, Zika empty vs Mock empty and the result contained in Supplementary Table 1. In the main text, the authors stated that 46 protein were enriched and 8 depleted in the membranes isolated from zika infected vs non-infected cells. Could the authors elaborate how they arrived at these numbers, please? Which statistical tests were performed? Which were the thresholds? Were all four comparisons used to arrive at the final list of 54 differentially regulated proteins?

Response: The statistical analysis was done using MSGLM R package dedicated to the Bayesian modeling of the MS data (a manuscript with the detailed description of the method is currently in preparation). In the original version of the manuscript, we were referring to (Stukalov et al. 2021), for the details of the analysis procedure. In the revised manuscript we have expanded the methods sections by explicitly providing these details. Our linear statistical model is using sparsity-inducing horseshoe+ prior for its effects. The p-values are directly calculated from the posterior distributions of the inferred effects by comparing them with zero. The significance thresholds for the effect sizes and p-values were indicated in the figure legends. We now also indicate these thresholds in the Supplementary Table 1 ("Legend" sheet).

Specifically, the protein groups highlighted in **Figure 2A** as significantly enriched in CNX-HA AP upon ZIKV infection have to satisfy all of the following criteria:

- Significantly enriched (median(\log_2 FC) ≥ 2 , p-value $\leq 1E-3$) in CNX-HA AP either in mock- or ZIKV-infected cells, where \log_2 FC is the difference between the \log_2 intensity of the protein group in CNX-HA AP and negative control AP conditions (median and p-value are taken over the posterior distribution of \log_2 FC).
- The CNX:ZIKV effect of the GLM model is significantly different from zero ($|\text{median}(\text{effect})| \geq 1$, two-sided p-value(effect) $\leq 1E-3$, where the p-value and median are taken over the posterior distribution of the effect). This effect corresponds to the following linear combination of \log_2 Intensities of 4 experimental conditions:
$$\text{effect}_{CNX:ZIKV} = (\log_2 I_{CNX,ZIKV} - \log_2 I_{CNX,Mock}) - (\log_2 I_{empty,ZIKV} - \log_2 I_{empty,Mock})$$
- The protein group should not contain contaminant or decoy proteins.

d. There is a vague information in the methods section about proteomics data processing, please include MaxQuant parameters used for the search and list all the data processing steps performed post MaxQuant.

Response: We thank this Reviewer to allow us to clarify our manuscript. We now have provided mqpar.xml as a part of our PRIDE submission PXD043372. It lists all the MaxQuant parameters. In the methods section we have specified that we did not change MaxQuant search parameters from their default values.

To provide full details of the post-MaxQuant data processing steps, we have now deposited our R and Julia scripts to github (<https://zenodo.org/record/8381245>). We have also expanded the methods section with a more detailed description of these steps.

e. Have any data imputation been done to LFQ values? If so, please explain how, in the methods section.

Response: To increase the depth and sensitivity of the MSGLM statistical analysis, it was done using MS1 precursor intensities (evidence.txt table of MaxQuant) directly as an input instead of LFQ values (similar to e.g. [msqrob2](https://www.bioconductor.org/packages/release/bioc/html/msqrob2.html) approach, <https://www.bioconductor.org/packages/release/bioc/html/msqrob2.html>). We did not impute intensity values before submitting them to the statistical analysis. Instead, the likelihood function of the statistical model accounted for missing values using probabilistic missing-not-completely-at-random model (similar to MSstats4 method (Kohler et al. 2023)). We have expanded the methods sections with a more detailed description.

d. The bait for affinity purification (CANX) should be highlighted in Fig2A. The authors should consider adding volcano plots for all 4 comparison as supplementary figures.

Response: We have now highlighted the CNX bait in green in the updated Fig. 2A. It is located in the top-right corner, as its log₂ fold-change enrichment over the background levels is above 5 both in mock- and in ZIKV-infected APs. We also provided the volcano plots as Supplementary Figure 1: one showing the hits enriched when comparing CNX-HA vs. empty cells for mock-infected cells, one showing the hits enriched when comparing CNX-HA vs. empty cells for ZIKV-infected cells, and, lastly, the volcano plot showing the changes in CNX-HA enrichment upon ZIKV infection (effect CNX:ZIKV). These plots cover the statistical tests we used to obtain the list of hits highlighted in Fig. 2A.

References

- Gokhale, N. S., A. B. R. McIntyre, M. D. Mattocks, C. L. Holley, H. M. Lazear, C. E. Mason, and S. M. Horner. 2020. 'Altered m(6)A Modification of Specific Cellular Transcripts Affects Flaviviridae Infection', *Mol Cell*, 77: 542-55 e8.
- Kohler, Devon, Mateusz Staniak, Tsung-Heng Tsai, Ting Huang, Nicholas Shulman, Oliver M. Bernhardt, Brendan X. MacLean, Alexey I. Nesvizhskii, Lukas Reiter, Eduard Sabido, Meena Choi, and Olga Vitek. 2023. 'MSstats Version 4.0: Statistical Analyses of Quantitative Mass Spectrometry-Based Proteomic Experiments with Chromatography-Based Quantification at Scale', *Journal of Proteome Research*, 22: 1466-82.
- Paul, D., S. Hoppe, G. Saher, J. Krijnse-Locker, and R. Bartenschlager. 2013. 'Morphological and biochemical characterization of the membranous hepatitis C virus replication compartment', *J Virol*, 87: 10612-27.
- Poet, G. J., O. B. Oka, M. van Lith, Z. Cao, P. J. Robinson, M. A. Pringle, E. S. Arner, and N. J. Bulleid. 2017. 'Cytosolic thioredoxin reductase 1 is required for correct disulfide formation in the ER', *EMBO J*, 36: 693-702.
- Stukalov, A., V. Girault, V. Grass, O. Karayel, V. Bergant, C. Urban, D. A. Haas, Y. Huang, L. Oubraham, A. Wang, M. S. Hamad, A. Piras, F. M. Hansen, M. C. Tanzer, I. Paron, L. Zinzula, T. Engleitner, M. Reinecke, T. M. Lavacca, R. Ehmann, R. Wolfel, J. Jores, B. Kuster, U. Protzer, R. Rad, J. Ziebuhr, V. Thiel, P. Scaturro, M. Mann, and A. Pichlmair. 2021. 'Multilevel proteomics reveals host perturbations by SARS-CoV-2 and SARS-CoV', *Nature*, 594: 246-52.
- Viedma-Poyatos, Á, P. González-Jiménez, O. Langlois, I. Company-Marín, C. M. Spickett, and D. Pérez-Sala. 2021. 'Protein Lipoxidation: Basic Concepts and Emerging Roles', *Antioxidants (Basel)*, 10.

REVIEWERS' COMMENTS

Reviewer #1 (Remarks to the Author):

The authors have appropriately addressed the concerns of this reviewer.

Reviewer #2 (Remarks to the Author):

The authors very nicely addressed all of my prior concerns. Good job.

Reviewer #3 (Remarks to the Author):

Thank you for the thorough responses and the additions made to the manuscript. I have no further comments.